

# Isolated sign language recognition through integrating pose data and motion history images

Ali Akdağ[1] and Ömer Kaan Baykan[2]

[1] Department of Computer Engineering, Tokat Gaziosmanpaşa University, Tokat, Turkey
[2] Department of Computer Engineering, Konya Technical University, Konya, Turkey

## ABSTRACT

This article presents an innovative approach for the task of isolated sign language recognition (SLR); this approach centers on the integration of pose data with motion history images (MHIs) derived from these data. Our research combines spatial information obtained from body, hand, and face poses with the comprehensive details provided by three-channel MHI data concerning the temporal dynamics of the sign. Particularly, our developed finger pose-based MHI (FP-MHI) feature significantly enhances the recognition success, capturing the nuances of finger movements and gestures, unlike existing approaches in SLR. This feature improves the accuracy and reliability of SLR systems by more accurately capturing the fine details and richness of sign language. Additionally, we enhance the overall model accuracy by predicting missing pose data through linear interpolation. Our study, based on the randomized leaky rectified linear unit (RReLU) enhanced ResNet-18 model, successfully handles the interaction between manual and non-manual features through the fusion of extracted features and classification with a support vector machine (SVM). This innovative integration demonstrates competitive and superior results compared to current methodologies in the field of SLR across various datasets, including BosphorusSign22k-general, BosphorusSign22k, LSA64, and GSL, in our experiments.

# INTRODUCTION

Deaf and hearing-impaired individuals primarily communicate through sign language, which comprises a variety of visual and kinesthetic components, including hand gestures, facial expressions, body movements, and posture. The language is utilized by a vast number of individuals worldwide and possesses an intricate linguistic structure that differs among regions and cultures (*Zahid et al., 2022*). Additionally, sign language is vital for the social integration of hearing-impaired individuals. Nevertheless, its prevalence among the general populace is limited, and there is an inadequate supply of sign language interpreters. Society as a whole is unable to effectively communicate with the hearing-impaired as a result. In order to address this issue, the disciplines of computer vision and machine learning are actively seeking novel approaches to autonomously transcribe sign language into textual or auditory formats. The primary objective of sign language recognition (SLR)

Corresponding author
Ali Akdağ, ali.akdag@gop.edu.tr

systems developed within this framework is to enhance the communication capabilities of individuals with hearing impairments, facilitating their interaction with individuals who lack proficiency in sign language.

Studies on SLR can be categorized into two primary groups, which are distinguished by the types of input data employed: sensor-based gloves and vision-based systems (*Ahmed et al., 2018*). The utilization of gloves equipped with flexion, proximity sensors, and accelerometers has been examined in a number of studies (*Oz & Leu, 2011*; *Gupta & Kumar, 2021*; *Liu et al., 2023*) focused on sensor-based glove research. One limitation of this approach is the requirement for the user to utilize an electronic glove equipped with multiple sensors positioned on the wrist or hand, thereby impeding the individual's ability to engage in a seamless and ergonomic human-computer interaction. Furthermore, this approach fails to acknowledge the non-manual gestural components that accompany sign language, including facial expressions, lip movements, eye movements, and body language (*Ibrahim, Zayed & Selim, 2019*). In the context of vision-based systems, the acquisition of essential data is facilitated by cameras, thereby obviating the requirement for sensors within the sensory glove. Additionally, these systems permit the assessment of non-manual characteristics that contribute to the formation of the sign. Due to these advantages and the broader applicability of vision-based systems, this study focuses on image-based systems.

From another perspective, SLR systems can be divided into two main categories based on the form of the input data used: static and dynamic. Static input typically consists of a single, still image of a sign, while dynamic inputs use video recordings or sequences of images of moving signs to capture the fluid and changing nature of sign language. This necessitates the evaluation of the sign both spatially and temporally. Dynamic inputs can be further subdivided into isolated (*Sarhan & Frintrop, 2023*) and continuous (*Aloysius & Geetha, 2020*) categories. Isolated input usually presents a single sign or word separately, with clear start and end points for each sign. On the other hand, continuous input encompasses the presentation of signs in a natural flow of sign language, with transitions between signs and the natural flow of language (*Rastgoo, Kiani & Escalera, 2021*). This research specifically targets the classification of isolated sign language words by employing a novel approach that utilizes pose data extracted from videos to capture spatial information and motion history images (MHIs) to analyze temporal dynamics, effectively integrating these aspects through the application of the ResNet-18 model.

Our research aims to address the following questions in the context of SLR system development:

1) What impact does the fusion of features extracted from ResNet-18 models trained on pose and MHI data have on the classification accuracy of SLR?
2) How do finger pose-based MHIs affect the accuracy of SLR?
3) What is the impact of the interaction of manual and non-manual features on the overall performance of SLR systems?
4) How does the estimation of missing pose data through linear interpolation affect the performance of models?

In pursuit of answers, we propose a system that integrates training the ResNet-18 model with pose data for spatial feature extraction and augments it with temporal information *via* MHIs. Recognizing the nuanced contributions of finger gestures, we introduce finger pose-based MHI (FP-MHI) to capture subtle gesture dynamics. We also explore the efficacy of linear interpolation in estimating missing pose data and assess the impact of including body and face data in the classification process. By fusing all extracted features and employing a support vector machine (SVM) for classification, our methodology demonstrates competitiveness with existing SLR approaches, highlighting the potential of our integrated approach to significantly enhance SLR.

Following the "Introduction", the study comprises the following sections: The "Related Studies" section summarizes existing methodologies in the field of SLR and the contributions of this study. The "Materials and Methods" section includes the details of the datasets used, the preprocessing steps applied, feature extraction, and classification techniques. The "Experimental Studies" section presents the classification results of the models, the effects of completing missing pose data, a comprehensive performance analysis of the proposed methodology, temporal and cost analyses, and comparisons with other studies. In the "Conclusion" section, the findings are evaluated, and recommendations for the future development of SLR systems are provided.

## RELATED STUDIES

The development of SLR technologies ranges from early versions, initially based on handcrafted features and classical machine learning techniques, to deep learning models that have revolutionized feature extraction and classification. This section aims to provide an overview of recent progress in isolated SLR and the methodologies used, with a particular focus on work using datasets such as BosphorusSign22k (*Camgoz et al., 2016*; *Özdemir et al., 2020*), LSA64 (*Ronchetti et al., 2016*), and GSL (*Adaloglou et al., 2020*).

Early versions of vision-based SLR systems relied mainly on handcrafted features and machine learning methods. Examples of these features are scale-invariant feature transform (SIFT) (*Yasir et al., 2016*), speeded-up robust features (SURF) (*Yang & Peng, 2014*; *Kour & Mathew, 2017*), histogram of oriented gradients (HOG) (*Raj & Jasuja, 2018*), and principal component analysis (PCA) (*Gweth, Plahl & Ney, 2012*; *Madana Mohana & Rama Mohan Reddy, 2014*), which are addressed by various machine learning algorithms for sign classification. However, these studies have predominantly focused on recognizing static signs such as letters and numbers. Transitioning from these foundational approaches, recent works have introduced more advanced techniques. For instance, *Madani & Nahvi (2013)* classified the extracted features with the minimum distance using the CAMSHIFT algorithm for tracking and the radon transform for feature extraction. This method was able to correctly identify 20 isolated dynamic Pakistani Sign Language words with 95.56% accuracy. *Li, Kao & Kuo (2016)* utilized the hidden Markov model (HMM) as a classifier to recognize 11 Taiwanese Sign Language words, integrating PCA with an entropy-based K-means algorithm for data transformation to eliminate redundant dimensions without losing essential information, and incorporated the ABC and Baum-Welch algorithms for optimization, achieving a recognition accuracy of 91.30%.

*Ahmed, Chanda & Mitra (2017)* introduced an image-based hand gesture recognition system that utilizes dynamic time warping (DTW). By monitoring hand movements and analyzing features of both the hands and face, the system achieved a 90% accuracy rate in recognizing 24 gestures from Indian Sign Language. *Ronchetti, Quiroga & Lanzarini (2016)* proposed a probabilistic model based on various features such as location, movement, and hand shape. This model achieved an accuracy of 97% on the LSA64 dataset (64 classes) and also showed an average accuracy of 91.7% on signer-independent tests. *Rodríguez & Martínez (2018)* designed a model based on the cumulative shape difference and SVM method and achieved an accuracy of 85% on the LSA64 dataset. *Özdemir et al. (2020)* employed the improved dense trajectories (IDT) method, which tracks and analyzes motion by capturing feature points across video frames and extracting features such as the histogram of oriented gradients (HOG), histogram of optical flow (HOF), and motion boundary histogram (MBH); they achieved an 88.53% accuracy on BosphorusSign22k (744 classes). Although these methods, which are based on traditional machine learning methods, have made significant strides in SLR, the results obtained are relatively low considering the number of words classified. This is because these methods cannot fully model the complex and dynamic nature of sign language. Furthermore, traditional machine learning algorithms often require complex feature engineering, which necessitates domain-specific expertise. This process requires the extraction of strong handcrafted features through detailed data analysis before the optimal features are selected to proceed with the machine learning algorithm. Moreover, as the diversity of the dataset increases, the development of new and more sophisticated methods may become necessary. Over time, with technological advancements and the evolution of machine learning, deep learning techniques have significantly reduced the need for manual feature engineering by automating the feature extraction process.

Deep learning is a machine learning method that consists of multiple layers that are capable of automatically learning data representations and modeling hierarchical levels of features (*LeCun, Bengio & Hinton, 2015*). Demand for deep learning technology increased after the 2012 ImageNet competition, when AlexNet, a convolutional neural network model developed by *Krizhevsky, Sutskever & Hinton (2012)*, reduced the error rate from 26.2% to 15.4% over a thousand categories. This groundbreaking development has allowed deep learning modeling to evolve rapidly thanks to high-performance GPUs developed and made available to process large datasets. Deep learning has achieved significant success in visual and auditory recognition tasks, such as classification, detection, segmentation, natural language processing, and other complex tasks (*LeCun, Bengio & Hinton, 2015*).

The innovations brought by deep learning models in the field of SLR have significantly improved the ability of systems to understand more complex and diverse signs. Existing techniques based on deep learning are generally based on convolutional neural networks (CNN). Sign language letters and digits are usually represented by static images, and 2D CNNs are sufficient to recognize these static signs (*Wadhawan & Kumar, 2020*; *Güney & Erkuş, 2022*; *Damaneh, Mohanna & Jafari, 2023*). However, the representation of words and sentences, by the very nature of the signs, has both spatial and temporal properties. This has led researchers to develop sophisticated methods to detect not only the spatial

location of movements and gestures but also the sequence and dynamics of these movements over time in order to extract the meaning of words and sentences.

Building on this foundation, the development and application of deep learning-based approaches have catalyzed profound advances in the field of SLR, marking a pivotal shift towards models that integrate both spatial and temporal dimensions. For example, *Masood et al. (2018)* designed a model that classifies by extracting temporal features by using spatial features obtained using a CNN as the input of an recurrent neural network (RNN). Their model achieved a 95.2% test accuracy on a subset of the LSA64 dataset (46 classes). *Zhang & Li (2019)* designed an artificial neural network called the multiple extraction and multiple prediction (MEMP) network, which combines a 3D CNN and convolutional long short-term memory (ConvLSTM) to extract and evaluate time- and space-based features of dynamic videos. The structure that they developed showed a test success of 99.063% on LSA64. *Imran & Raman (2020)* fine-tuned three pre-trained CNN models using various types of data that can represent a video in a single image, such as motion history images, dynamic images, and RGB motion image templates. The results of these three models were combined with an innovative fusion methodology using a kernel-based extreme learning machine, achieving an accuracy of 97.81% on LSA64. In the study by *Özdemir et al. (2020)*, the mixed convolution 3D residual networks (MC3) method, which has a structure combining both 2D and 3D convolutional layers, achieved a 78.85% accuracy on the BosphorusSign22k dataset. *Adaloglou et al. (2020)* used the GoogLeNet+TConvs (*Cui, Liu & Zhang, 2019*), 3D-ResNet (*Pu, Zhou & Li, 2019*), and I3D (*Szegedy et al., 2016*) models to classify Greek Sign Language (GSL) words consisting of 310 different classes. With these models, accuracy rates of 86.03%, 86.23%, and 89.74%, respectively, were achieved in signer-independent tests. *Wang et al. (2021)* proposed a (2+1)D convolution-based (2+1) D-SLR network that can achieve a higher accuracy at a faster rate. This network achieved a 98.7% accuracy on the LSA-64 dataset. *Sincan & Keles (2021)* trained the I3D model using RGB videos and trained the ResNet-50 model with RGB-MHI data. They achieved a 94.83% accuracy using the method in which they combined the features obtained from the models with the late fusion technique on the BosphorusSign22k dataset. Although these studies have achieved considerable success, they have not fully captured the complexity and diversity of sign language videos. In particular, there has been insufficient focus on the hands and facial expressions that convey much of the meaning in sign language. Facial expressions and gestures, as well as the detailed movements and positions of the hands, are critical elements that reinforce the meaning of sign language. The fact that existing studies ignore these important components has led to the search for new ways to further improve SLR systems. In this context, *Gökçe et al. (2020)* trained mixed convolutional 3D networks (MC3-18) using hand, body, and face images. By fusing the features obtained from these networks at the score level, they achieved a 94.94% accuracy on BosphorusSign22k test data. *Gündüz & Polat (2021)* used data on the face, hands, and full body regions and optical flow data in Inception3D model training and body and hand pose data in LSTM network training. The feature streams obtained from the trained models were combined and used as the input of a two-layer neural network, and a test accuracy of 89.3% was obtained with this method on BosphorusSign22k-general.

Despite the significant progress made in this work, SLR systems need to be more robust to background variation and signer variation. Furthermore, it is important to be able to perform a more detailed analysis to better understand the fine details of sign language. These needs emphasize the importance of systems based on pose data. Pose data have the potential to overcome these limitations when developing SLR systems. These data can reduce the impact of background variations and signer diversity on the overall performance of the systems. By directly capturing the positions and movements of the hands, face, and body, these data can enable systems to provide consistent results even under different conditions. This allows SLR systems to become more robust and reliable. In this context, *Konstantinidis, Dimitropoulos & Daras (2018b)* proposed a model that uses hand and body skeleton information extracted from RGB videos to train LSTM networks and combines the features obtained from these networks with the late fusion technique. Their model achieved an accuracy rate of 98.09% on LSA64. In another study (*Konstantinidis, Dimitropoulos & Daras, 2018a*), the same authors improved the performance of their method to 99.84% by adding additional streams analyzing RGB video and optical flow information. *Kindiroglu, Ozdemir & Akarun (2019)* achieved an accuracy rate of 81.58% in their studies on a subset of 174 classes from the BosphorusSign22k dataset by utilizing the temporal accumulative features (TAF) method. This method involves accumulating pose information from sign language videos and representing it with hue-based coloring. In a subsequent article, the same authors (*Kındıroglu, Özdemir & Akarun, 2023*) proposed a new method called "aligned temporal accumulated features" (ATAF), which is a pose-based visual representation to represent variable-length videos with fixed-length features, and achieved a 94.9% accuracy on the BosphorusSign22k dataset by combining this method with MC3-18 (*Tran et al., 2018*) models. *Selvaraj et al. (2021)* adopted an approach based on pose data using the LSTM, transformer, spatio-temporal graph convolution network (ST-GCN), and graph convolution network with structured learning (SL-GCN) models, which achieved accuracy rates of 86.6%, 89.5%, 93.5%, and 95.4%, respectively, in experiments on the GSL dataset. *Alyami, Luqman & Hammoudeh (2023)* classified hand and face pose data using a transformer-based model. On the LSA64 dataset, they achieved 98.25% and 91.01% accuracies in signer-dependent and signer-independent tests, respectively. *Özdemir, Baytaş & Akarun (2023)* designed a framework based on spatio-temporal graph convolution networks (ST-GCN) and multi-cue long short-term memory (MC-LSTM). This framework uses various gestural information such as body pose data and the face and hands to recognize sign language more accurately. This method achieved an accuracy of 92.58% in tests on the BosphorusSign22k dataset.

The above studies in the field of isolated SLR are summarized in Table 1. In this table, the reference, input modality, sign language parameter used, methodology, number of classes classified, and accuracy of the proposed method are given.

The review of related studies has demonstrated significant advancements in the area of isolated SLR; however, a gap still exists in the comprehensive analysis of fine motor movements and finger gestures. Current methodologies tend to focus predominantly on hand and body movements, while techniques that capture the subtle nuances of sign

**Table 1 Summary of related studies on isolated SLR.**

| Studies | Input modality | Params | Methodology | Number of classes | Accuracy (in %) |
|---|---|---|---|---|---|
| *Madani & Nahvi (2013)* | RGB | Full frame | CAMSHIFT, radon transform, minimum distance | 20 | 95.56 |
| *Ahmed, Chanda & Mitra (2017)* | RGB | Full frame | DTW | 24 | 90 |
| *Ronchetti et al. (2016)* | RGB | Hand | BoW+SubCls | 64 | 91.7 |
| *Rodríguez & Martínez (2018)* | RGB | Full frame | Cumulative SD-VLAD with SVM | 64 | 85 |
| *Özdemir et al. (2020)* | RGB | Full frame | MC3-18 | 744 | 78.85 |
| *Masood et al. (2018)* | RGB | Full frame | CNN, RNN | 46 | 95.2 |
| *Konstantinidis, Dimitropoulos & Daras (2018a)* | RGB, pose, optical flow | Body, hand, face | VGG-16, LSTM | 64 | 99.84 |
| *Konstantinidis, Dimitropoulos & Daras (2018b)* | Pose | Body, hands | LSTM | 64 | 98.09 |
| *Zhang & Li (2019)* | RGB | Full frame | MEMP network | 64 | 99.063 |
| *Kindiroglu, Ozdemir & Akarun (2019)* | Pose | Full frame | TAF & Hue subunit detection, CNN | 174 | 81.58 |
| *Imran & Raman (2020)* | MHI, dynamic image, RGBMI | Full frame | CNN, kernel-based extreme learning machine | 64 | 97.81 |
| *Özdemir et al. (2020)* | RGB | Full frame | IDT | 174 | 88.53 |
| *Adaloglou et al. (2020)* | RGB | Full frame | GoogLeNet, TConvs | 310 | 86.03 |
| *Adaloglou et al. (2020)* | RGB | Full frame | 3D-ResNet, BLSTM | 310 | 86.23 |
| *Adaloglou et al. (2020)* | RGB | Full frame | I3D, BLSTM | 310 | 89.74 |
| *Gökçe et al. (2020)* | RGB | Body, hand, face | MC3-18, score-level fusion | 744 | 94.94 |
| *Gündüz & Polat (2021)* | RGB, pose, optical flow | Body, hand, face | Inception 3D & LSTM-RNN Fusion | 174 | 89.35 |
| *Wang et al. (2021)* | RGB | Full frame | R(2+1)D-SLR | 64 | 98.7 |
| *Selvaraj et al. (2021)* | Pose | Full frame | LSTM | 310 | 86.60 |
| *Selvaraj et al. (2021)* | Pose | Full frame | Transformer | 310 | 89.50 |
| *Selvaraj et al. (2021)* | Pose | Full frame | ST-GCN | 310 | 93.50 |
| *Selvaraj et al. (2021)* | Pose | Full frame | SL-GCN | 310 | 95.40 |
| *Sincan & Keles (2021)* | RGB, MHI | Full frame | ResNet-50, I3D | 744 | 94.83 |
| *Kındıroglu, Özdemir & Akarun (2023)* | RGB, pose | Full frame | ATAF, TTN, MC3-18, Fusion | 744 | 94.90 |
| *Alyami, Luqman & Hammoudeh (2023)* | Pose | Hand, face | Transformer | 64 | 91.09 |
| *Özdemir, Baytaş & Akarun (2023)* | RGB, pose | Body, hand, face | ST-GCN, MC-LSTM | 744 | 92.58 |

language are noticeably lacking. This study aims to fill this void by introducing an MHI feature based on finger pose analysis. In addition to finger pose-based MHI (FP-MHI), we are developing an approach that incorporates data from hand, body, and face poses. However, existing solutions that encompass all components often have a high cost, as they typically need to address both spatial and temporal features. We use these data to train the ResNet-18 model, which captures only spatial information, in contrast to complex models that process both spatial and temporal information simultaneously. We compensate for

the lack of temporal information with three-channel MHIs derived from these pose data. By fusing features extracted from ResNet-18 models, which have a relatively simple architecture, and classifying them using a powerful classifier, the SVM, our proposed method aims to provide a more accurate and comprehensive approach to SLR studies.

The main contributions of this study are summarized below:

1) **Detailed pose estimation and completion of missing pose data:** MediaPipe Holistic is utilized for detailed pose data acquisition and the missing pose data problem is addressed with linear interpolation.

2) **Application of MHIs to pose data:** MHIs are applied to pose data, enabling a representation that captures both the spatial and dynamic aspects of sign language gestures.

3) **Finger pose-based MHI (FP-MHI) feature:** The FP-MHI feature is introduced; it provides a more granular and accurate representation of sign language gestures through the detailed analysis of finger movements.

4) **Enhanced ResNet-18:** The ResNet-18 model is improved with the randomized leaky rectified linear unit (RReLU) activation function and a detailed comparison with other efficient state-of-the-art models is conducted.

5) **Integrated SLR system:** A comprehensive SLR system that fuses pose data, MHIs, and FP-MHI features, based on the ResNet-18 and SVM methods, is developed, achieving a high accuracy.

6) **Temporal and cost analysis:** The proposed SLR system is analyzed in terms of time and cost, demonstrating its adaptability to real-time applications while maintaining a high accuracy and reducing computational resources.

7) **Comprehensive evaluation:** A thorough evaluation of the proposed methods across multiple datasets and a comparison with the existing literature are carried out.

## MATERIALS AND METHODS

In this section, we detail the basic materials and methodology used in the development of our SLR system. We provide detailed information about the datasets that form the basis of our research, the use of MediaPipe for pose estimation, the linear interpolation technique used to fill in missing data points, the generation of visual pose data, the MHIs that reflect the temporal traces of motion, the ResNet-18 model that plays a fundamental role in our deep learning approach, the SVM preferred for feature classification, and other processes. The diagram in Fig. 1 visually summarizes the overall structure of our proposed SLR system and the interaction between the main components. The following subsections provide a comprehensive account of our work, covering the materials and methodology on which our research is based, using a systematic approach.

### Datasets

In this study, multiple sign language datasets are used to comprehensively evaluate the effectiveness of the proposed method. Each dataset serves a unique purpose in our

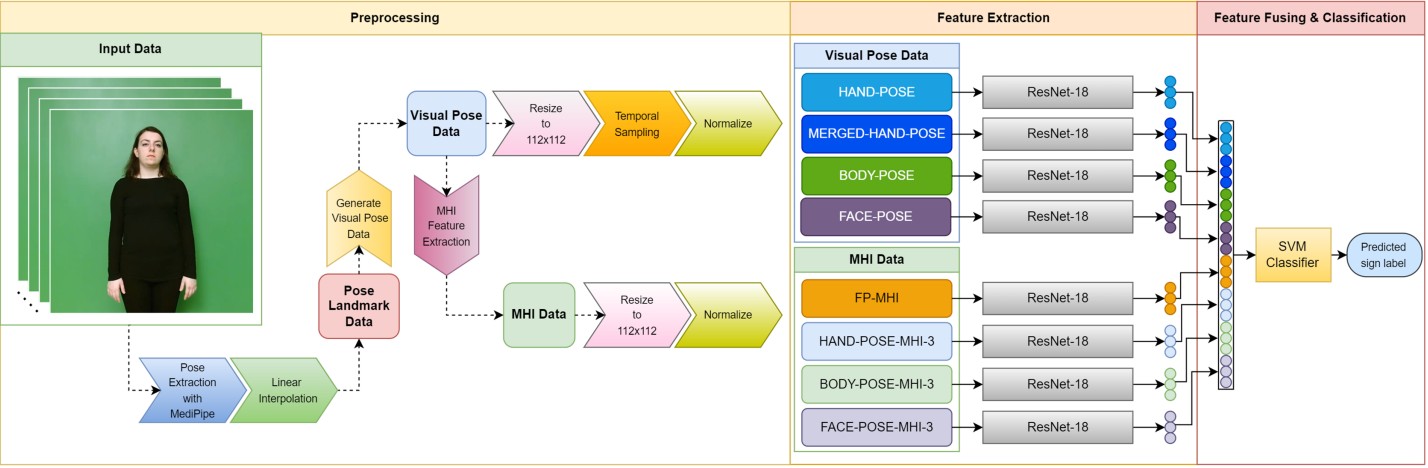

**Figure 1** Diagram of the proposed SLR system.

experimental design, from method development to a comparative analysis with works in the literature.

Our experimental analysis is based on BosphorusSign22k-general, a subset of the BosphorusSign22k dataset (*Camgoz et al., 2016*; *Özdemir et al., 2020*). This subset contains 4,839 training videos and 949 test videos for a total of 174 sign word classes performed by six signers. The data of one signer are reserved as test data for signer-independent evaluation. This general subset provides a diverse representation of Turkish Sign Language and provides an ideal platform for the initial testing and refinement of our proposed method.

The complete BosphorusSign22k dataset contains 22,542 videos and 744 sign word classes. The test set is reserved for signer-independent evaluation. This dataset is used to evaluate the scalability of our method and its effectiveness on a larger set of sign words. Thus, the ability of the proposed method to understand sign languages in different contexts is tested in a wider framework.

In order to verify the applicability of our proposed method across different sign languages, we also consider the LSA-64 dataset, which contains 64 sign words from Argentine Sign Language. This dataset, created by *Ronchetti, Quiroga & Lanzarini (2016)*, contains a total of 3,200 video images, each consisting of 10 signers repeating each word five times. The creators of the dataset did not specifically separate this dataset into training and testing sets. Therefore, based on studies in the literature, three different experimental sets were created in our study: Experiment 1 (E1), where the 5th and 10th signers are used as testing data; Experiment 2 (E2), where nine out of 10 signers are used for training data and the rest for testing data; and Experiment 3 (E3), where the dataset is randomly split into training data (80%) and testing data (20%).

Another dataset evaluated in our study is the GSL dataset, consisting of 310 Greek Sign Language words and created by *Adaloglou et al. (2020)*. This dataset consists of a total of 40,826 videos generated by seven different signers. The data of one signer were

**Table 2 Datasets used in the study.**

| Dataset | Reference | Training data | Test data | Number of signer | Class |
|---|---|---|---|---|---|
| BosphorusSign22k-General | *Camgoz et al. (2016)*, *Özdemir et al. (2020)* | 4,839 | 949 | 6 | 174 |
| BosphorusSign22k | *Camgoz et al. (2016)*, *Özdemir et al. (2020)* | 18,018 | 4,524 | 6 | 744 |
| LSA64 (E1) | *Ronchetti, Quiroga & Lanzarini (2016)* | 2,560 | 640 | 10 | 64 |
| LSA64 (E2) | *Ronchetti, Quiroga & Lanzarini (2016)* | 2,880 | 320 | 10 | 64 |
| LSA64 (E3) | *Ronchetti, Quiroga & Lanzarini (2016)* | 2,880 | 320 | 10 | 64 |
| GSL | *Adaloglou et al. (2020)* | 34,995 | 3,500 | 7 | 310 |

excluded from the training data for signer-independent evaluations. Of the data excluded from the training data, 2,331 videos are reserved as validation data and 3,500 are reserved as test data.

Using these various datasets, we aim to provide a comprehensive validation of our proposed method, highlighting its robustness, versatility, and scalability and its potential for wide applicability in SLR and understanding tasks. A summary of the datasets used in our study is given in Table 2.

## Pose extraction with MediaPipe

In the rapidly advancing fields of computer vision and machine learning, accurate human pose extraction from videos is crucial for applications such as SLR. A leading framework, MediaPipe's Holistic solution (*Lugaresi et al., 2019*; *Grishchenko & Bazarevsky, 2020*), provides a comprehensive solution to efficiently track face, hand, and body positions for real-time, accurate pose estimation using deep learning. Our work is seamlessly integrated into our research for improved pose inference by leveraging MediaPipe's Holistic solution. In this study, the reasons for choosing MediaPipe include its real-time processing capability, high accuracy rates, and extensive pose estimation features. MediaPipe effectively addresses the nuanced requirements encountered in SLR systems, thereby offering a high performance and low latency. Additionally, the fact that it is open-source, supported by a broad developer community, and continuously updated enhances the flexibility and adaptability of this technology.

In our study, first, using the MediaPipe Holistic solution, body, hand, and face pose landmark points were obtained from each video frame. A video (V) consists of N frames, denoted as $(I_1, I_2, \ldots, I_N)$. Each frame has dimensions $W \times H \times C$. W and H are the width and height of the frame, respectively, and C is the number of color channels. The MediaPipe Holistic model, denoted as M, processes each frame to output a set of landmarks $L_n$ for the face ($L_{face, n}$), hands ($L_{hands, n}$), and body ($L_{body, n}$) (Eq. (1)):

$$L_n = M(I_n) = \left\{ L_{face,n}, L_{hands,n}, L_{body,n} \right\}. \tag{1}$$

Here, $L_{face,n}$ contains 468 landmark points, $L_{hands,n}$ contains 42 landmark points (21 for each hand), and $L_{body,n}$ contains 33 landmark points. Each landmark point $l \in L_n$ is represented as a triple $(x, y, z)$, where x and y are the pixel coordinates of the landmark in

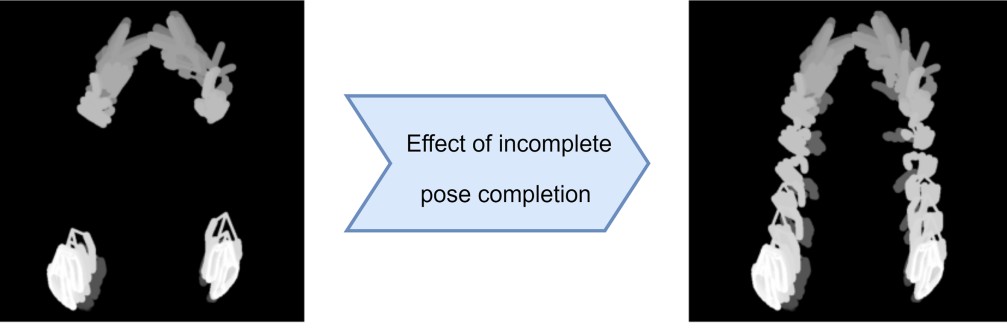

**Figure 2 Effect of incomplete pose completion on HAND-POSE-MHI-1.**

the image plane and z is the depth information relative to the camera plane. In the next subsection, the estimation of missing landmark points using linear interpolation and the generation of visual pose data from these points will be analyzed.

## Completion of missing pose data with linear interpolation

If the hand is not in an appropriate position relative to the camera angle, if it moves out of the camera's field of view, if the movements in the video are too fast, or if the hand is partially covered by an object or body during the video, MediaPipe may not be able to obtain the hand landmark points in some frames. Indeed, in the experiments with BosphorusSign22k-general, the dataset consists of 492,541 frames. In 21,551 of these frames, the right-hand pose landmark points were missing, and in 46,088 frames, the left-hand pose landmark points were missing. These missing landmark points may affect the classification performance.

In this study, we used linear interpolation to estimate missing hand pose data. Linear interpolation is a method used to estimate a new point between two known points. In the context of hand landmark data, imagine that there are n unknown hand landmark points between two known hand landmarks, and let i be the index of one of these unknown points. The value of a missing hand pose point, $L_{hand,missing,\,n}$, can be estimated using Eq. (2):

$$L_{hand,missing,\,n} = L_{hand,\,known,n-1} + i.\frac{\left(L_{hand,\,known,n+1} - L_{hand,\,known,n-1}\right)}{(n+1)}. \tag{2}$$

This formula allows us to interpolate the position of missing hand pose landmarks linearly by using the known positions of the hand landmarks immediately before and after the missing data, denoted as $L_{hand,\,known,n-1}$ and $L_{hand,\,known,\,n+1}$. As an example, the HAND-POSE-MHI-1 data obtained from the incomplete pose data and the HAND-POSE-MHI-1 data obtained from the completed pose data are shown in Fig. 2.

## Generation of visual pose data

After obtaining landmark points for the body, hands, and face from each video frame using the MediaPipe Holistic solution, visual pose data were created using the steps described in

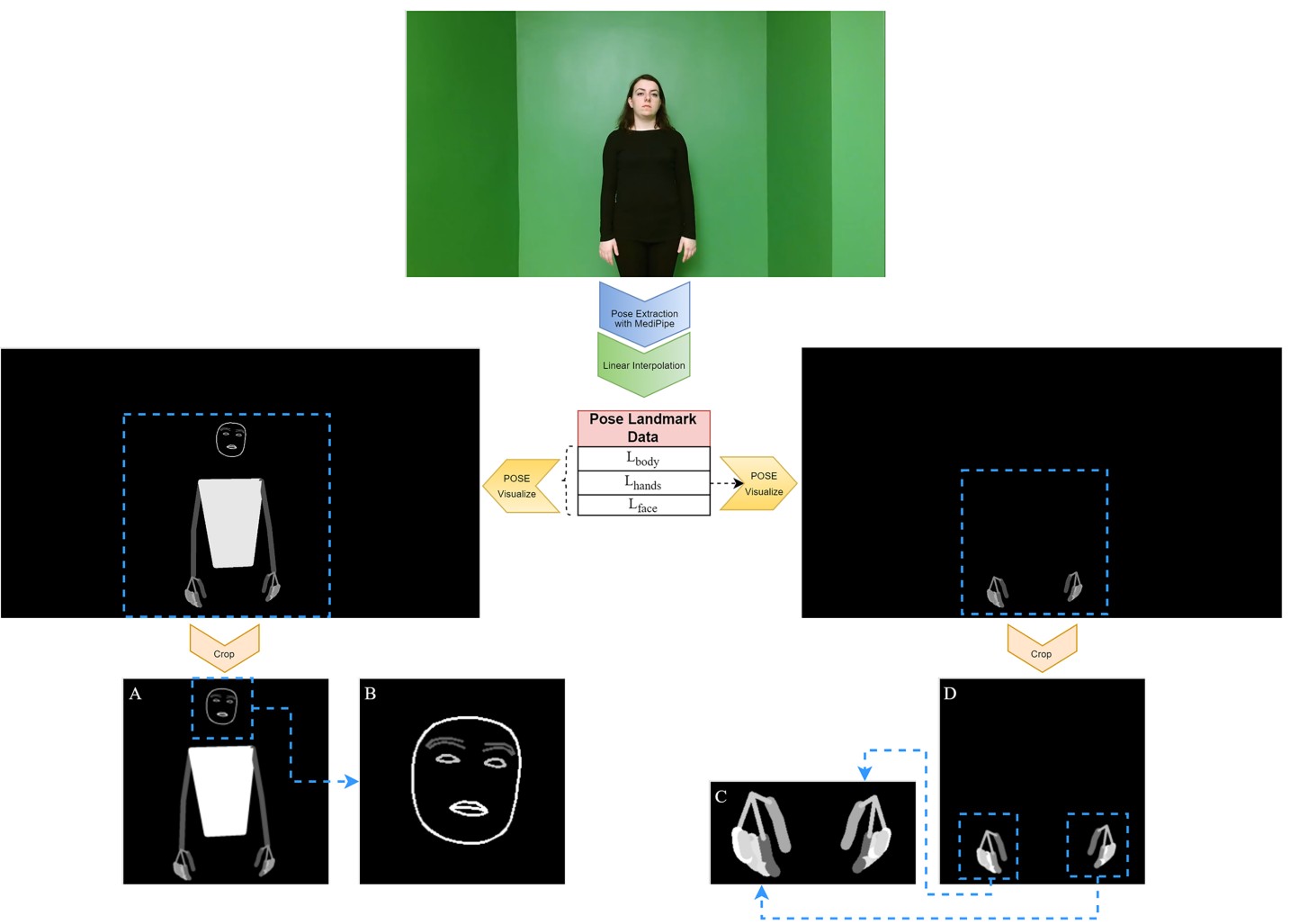

**Figure 3 Obtaining visual pose data from a frame.** (A) BODY-POSE, (B) FACE-POSE, (C) MERGED-HAND-POSE, (D) HAND-POSE.

this subsection. Converting numerical landmark points into visual representations improves the interpretability of pose estimation in areas such as SLR, allowing for a more intuitive analysis of the captured poses. This visual pose data are shown in Fig. 3.

First, the BODY-POSE image was created. These images provide a comprehensive representation that visualizes the body, hands, and face in a single frame. This process is performed using all landmark points $(L_n)$ and the links between them for each frame $(I_n)$ of the video (V). First, for each frame $(I_n)$, a blank black canvas $(C_n)$ of the same dimensions as the original image $(W \times H)$ is created. This canvas serves as a background for drawing both the landmark points and the connections between them. For each landmark point $l \in L$, a landmark point can be defined as $l = (x, y, z)$. Here, x and y denote the pixel coordinates of the landmark point on the canvas. We will not use the depth information z here, so we express a landmark point as $l = (x, y)$. The links between each

landmark point represent the relationships between specific landmark pairs. Each link can be assigned a different color and thickness value. The drawing of the links is performed by drawing lines between the start and end landmark points. The drawing is done for each $link_k$, $link_k = (l_{start}, l_{end})$, using the start and end points defined as $l_{start} = (x_{start}, y_{start})$ and $l_{end} = (x_{end}, y_{end})$. The drawing function is given below (Eq. (3)):

$$DrawLine(C_n, l_{start}, l_{end}, color_k, thickness_k). \tag{3}$$

It draws a line on the canvas ($C_n$) between the points $l_{start}$ and $l_{end}$, with the specified $color_k$ and $thickness_k$. This function provides a visual representation of both the points and the connections between them and is the basis of the visualization process.

In addition to BODY-POSE images, our study focused on the detailed analysis of hand gestures, a critical component of sign language. For this purpose, HAND-POSE images were produced, showing only hand poses. These images were drawn using only the landmark points ($L_{hands}$), with steps similar to the BODY-POSE generation steps.

After obtaining the BODY-POSE and HAND-POSE data, focus area cropping was performed on these images. This process aims to create a standardized structure to be used in subsequent analysis phases by cropping the unnecessary parts of the visual data. First, we identify the bounding boxes for the data to be cropped. For these bounding boxes, we use a set of $L_n$ landmark points obtained over the whole video. For the given set of $L_n$ landmark points, that is, for each landmark point $l_i = (x_i, y_i)$, the process of finding the minimum and maximum values of x and y between $i = 1, 2, \ldots, N$ is mathematically shown in Eqs. (4)–(7):

$$x_{min} = \min_{i=1}^{N} x_i, \tag{4}$$
$$y_{min} = \min_{i=1}^{N} y_i, \tag{5}$$
$$x_{max} = \max_{i=1}^{N} x_i, \tag{6}$$
$$y_{max} = \max_{i=1}^{N} y_i. \tag{7}$$

After finding the minimum and maximum values of the x and y coordinates, we add a padding value to leave enough space for the landmark points and their connections to be fully visible in the image. During this process, we check that the minimum values are not less than 0 and that the maximum values are not greater than the height (H) and width (W) of the image. These corrections guarantee that the bounding box does not exceed the area of the image and contains all relevant landmarks. The operations are shown in Eqs. (8)–(11):

$$x'_{min} = \max(0, x_{min} - padding), \tag{8}$$
$$y'_{min} = \max(0, y_{min} - padding), \tag{9}$$
$$x'_{max} = \min(W, x_{max} + padding), \tag{10}$$
$$y'_{max} = \min(H, y_{max} + padding). \tag{11}$$

After this process, we make the bounding box approach the square format. To do this, we first find the width (W′) and height (H′) of the bounding box, as shown in Eqs. (12) and (13):

$$W' = x'_{max} - x'_{min}, \qquad (12)$$
$$H' = y'_{max} - y'_{min}. \qquad (13)$$

If the width of the bounding box is greater than its height, we need to expand the box around the height axis; otherwise, we need to expand the box around the width axis. The amount of expansion (Δ) we will use for this operation is calculated using Eq. (14):

$$\Delta = \frac{|W' - H'|}{2}. \qquad (14)$$

Once the expansion amount Δ is found, Eq. (15) is used to expand the bounding region:

$$\begin{cases} y'_{min} = \max(0, y_{min} - \Delta), & y'_{max} = \min(H, y_{max} + \Delta) & \text{if } W' > H', \\ x'_{min} = \max(0, x_{min} - \Delta), & x'_{max} = \min(W, x_{max} + \Delta) & \text{otherwise.} \end{cases} \qquad (15)$$

This equation provides the coordinate information necessary to proportionally adjust both the width and length of the bounding region without exceeding the physical boundaries of the image. The cropping process shown in Eq. (16) is then performed for each frame of the video (V):

$$I'_n = I_n \left[ y'_{min} : y'_{max}, x'_{min} : x'_{max} \right]. \qquad (16)$$

when applied to each frame of the video (V), this process ensures that all frames in the video are cropped in a coherent and focused manner. As a result, this process focuses on the area of the pose images in the original image and eliminates redundant data. When applied to each frame of the video (V), this process ensures that all frames within the video are cropped in a coherent and focused manner. The result is a focus on the area of the pose images in the original image and the elimination of redundant data.

After obtaining the BODY-POSE and HAND-POSE data, two new features were created using these data: FACE-POSE and MERGED-HAND-POSE. FACE-POSE is obtained by cropping the face region from the BODY-POSE data with the help of $L_{face}$ landmark points. MERGED-HAND-POSE is obtained from the HAND-POSE data by cropping the regions corresponding to the hands through $L_{hands}$ landmark points and then merging them. These cropped features provide a closer look and deeper insight into the subtleties of human poses. While the creation of FACE-POSE data is important for the focused analysis of facial expressions and nuances, similarly, MERGED-HAND-POSE data provide a richer context for the detailed interpretation of hand gestures and positions. These features are expected to influence the recognition accuracy in the SLR system.

## Motion history image

A motion history image (MHI) is a representation method used to summarize motion information from video frames in a single image (*Bobick & Davis, 2001*). MHIs are widely

used in motion analysis and motion-based classification applications. They are especially important in areas such as human interactions, gestures, and SLR (*Zhang et al., 2019*; *Naeem et al., 2020*; *Chandragiri & Ijjina, 2021*; *Sincan & Keles, 2021*; *Ghosh et al., 2023*).

The fundamental concept underlying MHIs is to visually represent the temporal progression of motion within a sequence of video frames. A weighted combination of every frame from the start to the end of the motion is required for this. The outcome of the MHI algorithm is an image, where the intensity of each pixel is calculated according to the temporal proximity of the motion at that pixel (*Ahad, 2013*).

The calculation of the MHI starts by taking consecutive differences of video frames. These differences indicate where and when the motion occurred. Then, this motion information is displayed in an MHI that accumulates across all frames. The MHI not only provides information about the location and time of the movement but also captures features such as the direction and speed of the movement. This is useful in situations where the dynamic properties of motion are important for classification (*Ahad et al., 2012*).

The mathematical formula of the MHI is expressed in Eq. (17):

$$H_\tau(x, y, t) = \begin{cases} \tau & \text{if } \psi(x, y, t) = 1, \\ \max(0, \ H_\tau(x, y, t-1) - \delta) & \text{otherwise.} \end{cases} \quad (17)$$

In this formula, $(x, y)$ represents pixel positions, and $t$ represents time. The parameter $\tau$ stands for the travel time of a frame, and $\delta$ is called the decay parameter. $\psi$ is the update function describing the presence of motion. This update function (Eq. (18)) is applied every time a new video frame is analyzed:

$$\psi(x, y, t) = \begin{cases} 1 & \text{if } D(x, y, t) > \xi, \\ 0 & \text{otherwise,} \end{cases} \quad (18)$$

where $\xi$ is a threshold parameter. $D(x, y, t)$ represents the difference between two consecutive frames, which is defined in Eq. (19):

$$D(x, y, t) = |I(x, y, t) - I(x, y, t-1)|. \quad (19)$$

As a result of this process, pixels that have recently moved create a brighter image, while older movements create a darker image. As a result, the MHI is a grayscale image that visualizes motion in a video sequence.

Beyond the grayscale MHI, the RGB-MHI (*Sincan & Keles, 2021*) builds on the foundations of MHIs, extending the traditional MHI to provide a richer and more detailed representation of motion dynamics over time. An RGB-MHI is constructed by dividing the video sequence into three equal parts and computing separate motion histories for each. Each color channel—red, green, and blue—is assigned to represent motion information at specific time intervals. The calculation of the RGB-MHI can be considered an extension of the MHI calculation and is adapted to process and compile motion data across RGB channels.

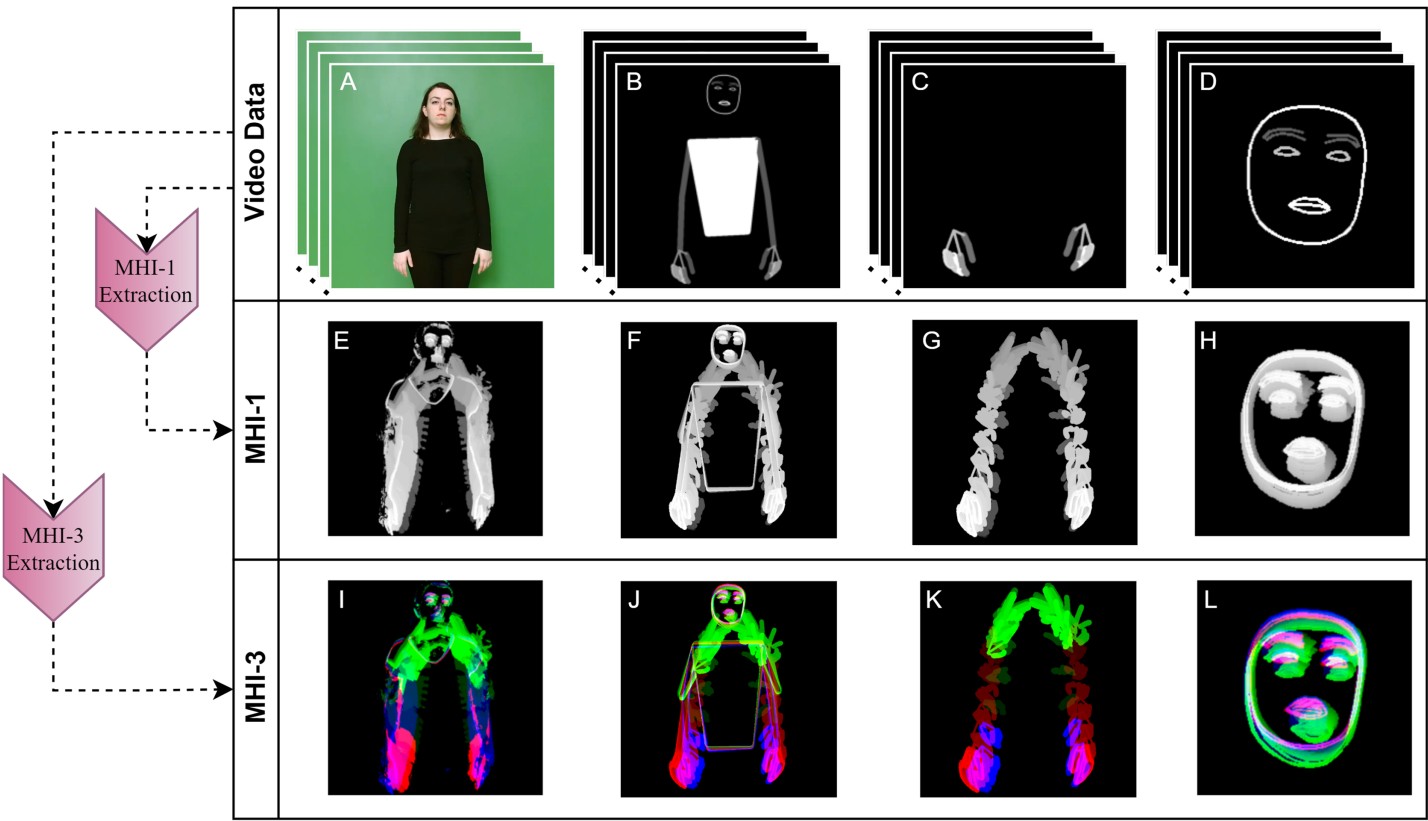

**Figure 4** **Extraction of MHI-1 and MHI-3 features of raw video and pose data.** (A) RAW, (B) BODY-POSE, (C) HAND-POSE, (D) FACE-POSE, (E) RAW-MHI-1, (F) BODY-POSE-MHI-1, (G) HAND-POSE-MHI-1, (H) FACE-POSE-MHI-1, (I) RAW-MHI-3, (J) BODY-POSE-MHI-3, (K) HAND-POSE-MHI-3, (L) FACE-POSE-MHI-3.

In this work, we apply the MHI and RGB-MHI methods to the BODY-POSE, HAND-POSE, and FACE-POSE data obtained in the previous subsection. The normal MHI and RGB-MHI are called MHI-1 and MHI-3, respectively, to indicate the number of channels. For a comparative analysis with these generated MHI data, we also extract MHI-1 and MHI-3 features from the RAW video data. The MHI-1 and MHI-3 features of an example sign are shown in Fig. 4.

**Finger pose-based MHI**

This subsection elucidates the development of the finger pose-based motion history image (FP-MHI) feature, a novel contribution of our research aimed at capturing the temporal dynamics of finger movements within SLR systems. Initially, visual pose data for each finger are meticulously generated by employing landmarks specific to each finger within $L_{hands,n}$, following the procedures outlined from Eqs. (3) to (16). This methodological approach results in the creation of five distinct image sequences, each providing a depiction of finger movements across a video frame. Subsequently, the MHI technique is individually applied to these sequences, yielding a dedicated MHI dataset for each finger.

Integrating these datasets culminates in the formation of a comprehensive five-channel FP-MHI image, encapsulating the essence of finger motion dynamics. The mathematical formulation of the FP-MHI is delineated in Eq. (20):

$$FP - MHI = Concat\left(H_\tau^{THUMB}(x, y, t), H_\tau^{INDEX}(x, y, t), H_\tau^{MIDDLE}(x, y, t), H_\tau^{RING}(x, y, t), H_\tau^{PINKY}(x, y, t)\right). \quad (20)$$

In this equation, for each finger, the MHI, denoted as $H_\tau^f(x, y, t)$, represents the accumulation over time (t) of the motion of the finger f at a given x, y pixel location, and the Concat function represents the process of merging the given MHIs, resulting in a 5×x×y FP-MHI that presents the accumulation over time of the motions of five different fingers in five different channels. In sign language classification, this FP-MHI is a crucial step in capturing more subtle gestures and meanings. This is an approach to the detailed analysis of finger gestures that has not been previously discussed in the literature. By employing this approach, the temporal progression of gesture data can be represented with greater accuracy, thereby potentially enhancing the precision of sign language classification. An example of the generation of FP-MHI data is shown in Fig. 5.

## Other preprocessing steps

A sample representation of the data that will be used for our final SLR system based on the processes described above is shown in Fig. 6. These data undergo a further series of preprocessing steps before they are given as input to the ResNet-18 models.

First, in order for the deep learning model used in our research to train efficiently and make accurate predictions, the input data must have a certain format and size. For this reason, all data were reduced to a 112 × 112 resolution, which is the size expected by the model as input. However, since the MERGED-HAND-POSE data contain the combined image of both hands, the size was set to 56 × 112.

Second, visual pose data are image sequences consisting of multiple frames that capture the evolution of motion over time. Therefore, for our ResNet-18 model to process these data effectively, we need to select a certain number of frames from each video sequence that the model can process. To increase the diversity and generalizability of the model training, we randomly select 32 frames from each video sequence in each training cycle. In this way, the model is able to learn the motions from different time periods and recognize the general motion pattern. However, in order to ensure temporal consistency and comparability in our dataset, we have selected equally spaced frames from each video sequence to evaluate and test the performance of our model. This method allows us to more fairly and consistently evaluate the performance of the model across different videos.

In the last stage, the normalization process was performed. This process is formulated in Eq. (21):

$$Normalized(x) = \frac{x - \mu}{\sigma}. \quad (21)$$

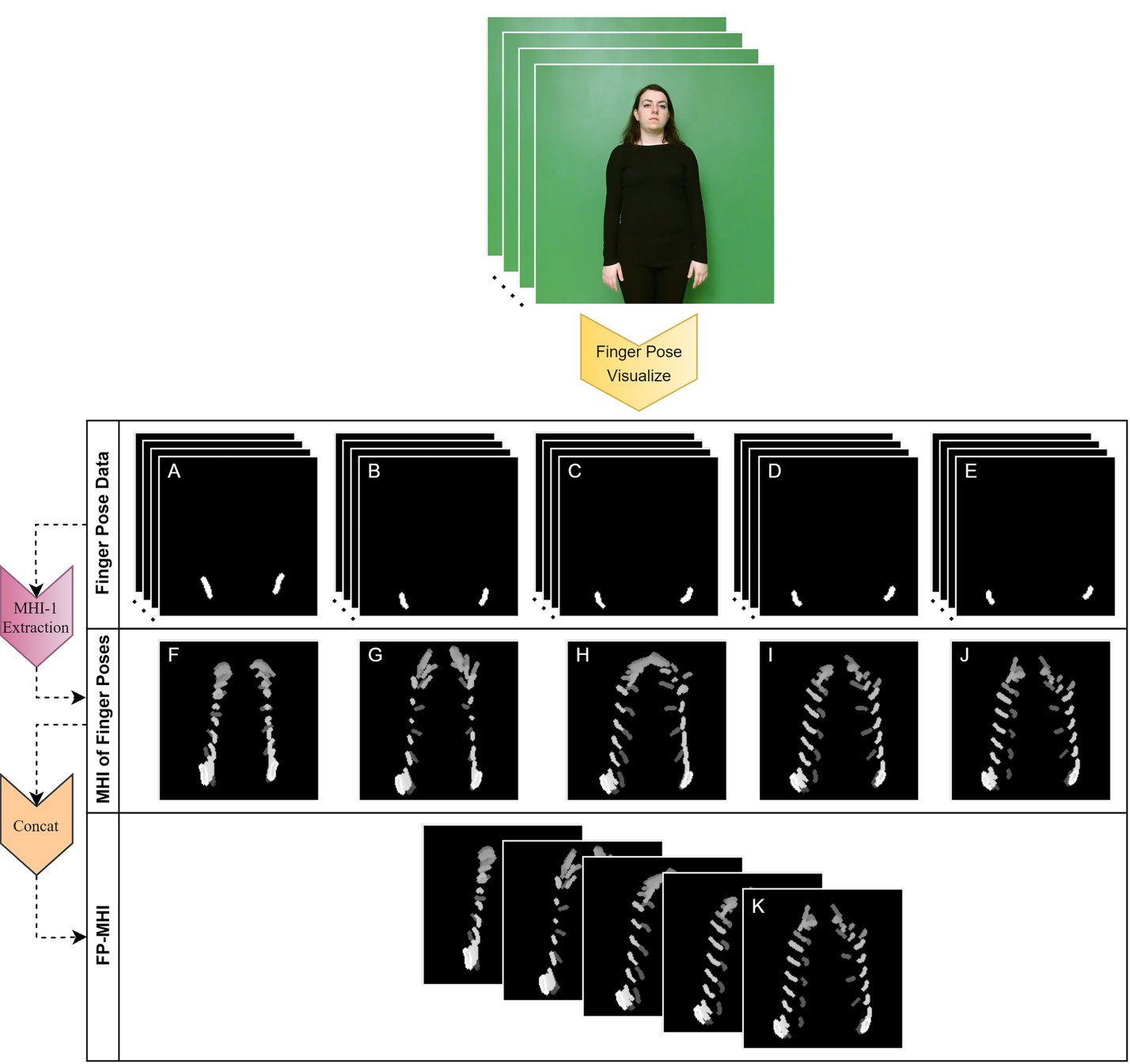

**Figure 5 Extracting the FP-MHI feature from finger pose data.** (A) THUMB POSE, (B) INDEX FINGER POSE, (C) MIDDLE FINGER POSE, (D) RING FINGER POSE, (E) PINKY FINGER POSE, (F) THUMB POSE-MHI, (G) INDEX FINGER POSE-MHI, (H) MIDDLE FINGER POSE-MHI, (I) RING FINGER POSE-MHI, (J) PINKY FINGER POSE-MHI, (K) FINGER-POSE-MHI (FP-MHI).

Here, x is the original pixel value, μ represents the mean, and σ represents the standard deviation. The normalization is performed to scale the input data and allow these data to be processed more efficiently by the model.

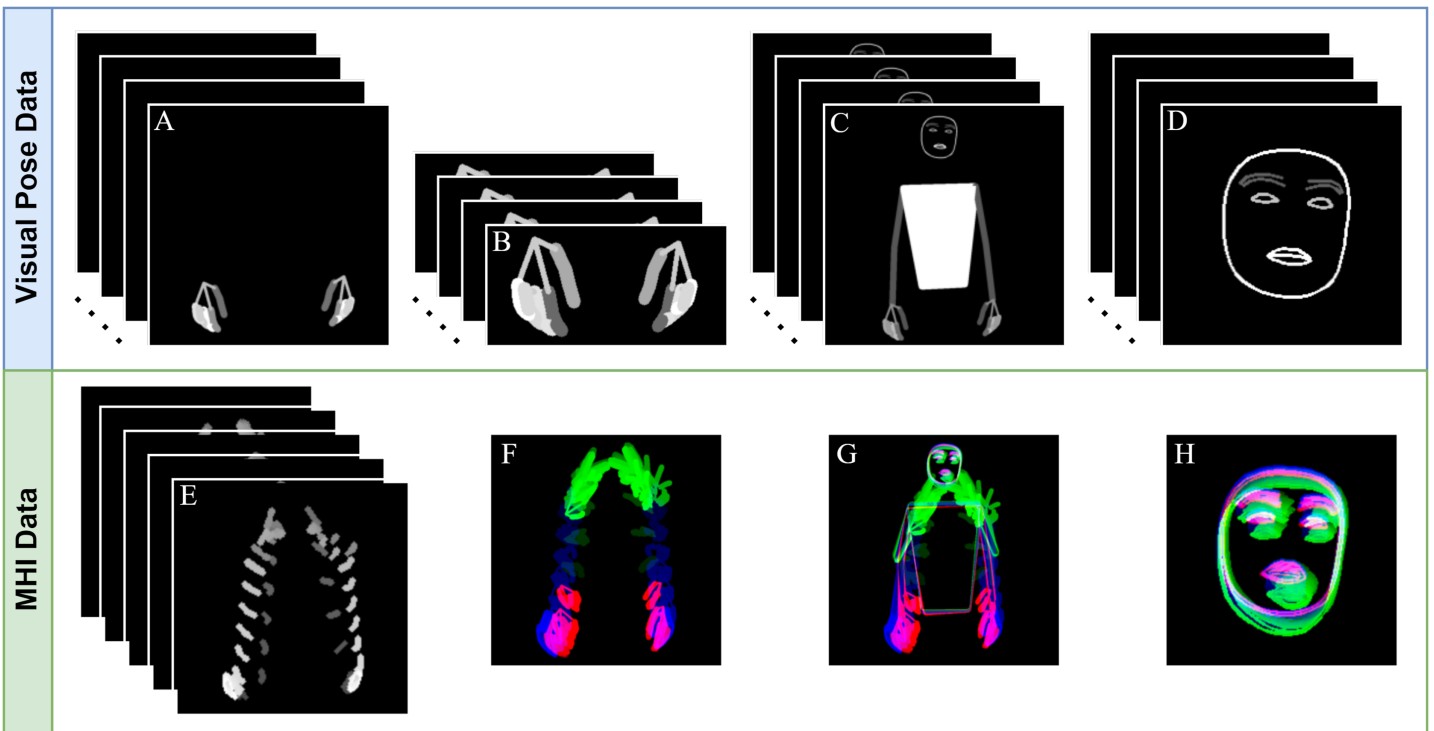

**Figure 6  Data to be used in the final SLR system.** (A) HAND-POSE, (B) MERGED-HAND-POSE, (C) BODY-POSE, (D) FACE-POSE, (E) FP-MHI, (F) HAND-POSE-MHI-3, (G) BODY-POSE-MHI-3, (D) FACE-POSE-MHI-3.

**Table 3  Data obtained after all preprocessing steps.**

| Data | Size |
| --- | --- |
| HAND-POSE | $32 \times 112 \times 112$ |
| MERGED-HAND-POSE | $32 \times 56 \times 112$ |
| BODY-POSE | $32 \times 112 \times 112$ |
| FACE-POSE | $32 \times 112 \times 112$ |
| FP-MHI | $5 \times 112 \times 112$ |
| HAND-POSE-MHI-3 | $3 \times 112 \times 112$ |
| BODY-POSE-MHI-3 | $3 \times 112 \times 112$ |
| FACE-POSE-MHI-3 | $3 \times 112 \times 112$ |

The data sizes after all preprocessing steps to be given as input to the ResNet-18 models for our final SLR system are shown in Table 3.

### ResNet-18 model

In our study, the ResNet-18 model (*He et al., 2016*) has been selected as the foundational model for the SLR system. This choice was made considering ResNet-18's ability to provide effective training in a short time and due to the fact that its structural features require relatively low computational costs. The skip connection architecture of the model

reduces the vanishing gradient problem encountered in training deeper networks while enhancing the model's learning capacity. This feature is particularly valuable when working with limited datasets, as it strengthens the model's ability to generalize from less data (*Oyedotun, Ismaeil & Aouada, 2023*).

During the adaptation of ResNet-18 for the SLR task, the architecture of the model was fine-tuned using pre-defined weights that had been trained on the ImageNet dataset. The transfer learning approach allows the model to quickly adapt to our sign language dataset, significantly speeding up the training process (*Shaha & Pawar, 2018*). This methodology aims to improve the performance in the new task, thereby enhancing the accuracy of sign language classification.

The structure of the model starts with a convolution layer with 64 filters with three channels (red, green, blue) for RGB images in the input layer. However, the data used in this study can have different sizes when given as input to our model: one channel (1D) for MHI-1, three channels (3D) for MHI-3 (RGB-MHI), five channels (5D) for finger pose MHI, and thirty-two channels (32D) for video pose data. Since the dimensions of these data are not compatible with the default input dimensions of ResNet-18, an appropriate preprocessing step was performed. Specifically, a convolution layer was added at the beginning of the model architecture with an input dimension varying depending on the type of data, an output dimension of 3, a 3 × 3 kernel size, and a step size of 1. This preprocessing step is of critical importance, as it transforms data of various sizes into a standard format that can be effectively processed by the ResNet-18 model. Such a transformation is essential for maintaining the standard dimensions of the data without hindering the model's ability to leverage its pre-trained weights for feature extraction.

Following the adapted input layer, the model incorporates a maximum pooling layer that effectively reduces the size of each feature map. Subsequently, four layers that constitute the model's depth are stacked, with each layer containing two residual blocks. These residual blocks assist in mitigating potential performance degradation when deeper architectures of the model are trained.

By default, the rectified linear unit (ReLU) activation function is used in the ResNet-18 model architecture. The formula of this activation function is shown in Eq. (22):

$$\text{ReLU}(x) = \max(0, x). \tag{22}$$

However, in the context of this study, the default activation function in the ResNet-18 architecture, ReLU, has been replaced with the randomized leaky rectified linear unit (RReLU). RReLU introduces non-parametric behavior by randomly selecting a slope for negative activation values within predefined lower and upper bounds. This approach enhances the model's capacity to learn and generalize, significantly improving its robustness against overfitting (*Xu et al., 2015*). The RReLU activation function can be mathematically represented as follows:

$$\text{RReLU}(x) = \begin{cases} x & \text{if } x \geq 0, \\ ax & \text{otherwise}. \end{cases} \tag{23}$$

In this formula, a is a coefficient that is randomly drawn from a uniform distribution between the specified lower (default:1/8) and upper (default:1/3) bounds during the training phase. This randomness introduces variability into the activation, which contributes to the model's ability to generalize better and prevent overfitting (*Banerjee, Mukherjee & Pasiliao, 2020*). In the "Experimental Studies" section, the effect of using the RReLU activation function instead of ReLU is analyzed.

### Fusing of extracted features

In order to perform the SLR task in our study, a separate pre-trained ResNet-18 model was trained for each feature obtained from the preprocessing stage, and after the training processes were completed, the features obtained by from the flattening layers of models were extracted to be used in the fusion and classification processes.

The fusion of features was accomplished through a concatenation operation, a method chosen for its simplicity and effectiveness in preserving the integrity and distinctiveness of each feature set. Suppose that we have n feature vectors $v_1, v_2, \ldots, v_n$, each with a dimensionality of 512, extracted from the flattening layers of the pre-trained ResNet-18 models. The concatenated feature vector $v_{fused}$ is then formulated as follows (Eq. (24)):

$$V_{fused} = \text{Concat}(v_1, v_2, \ldots, v_n). \tag{24}$$

This concatenated vector $V_{fused}$ amalgamates the individual strengths of each feature set, thereby providing a holistic representation of the different gesture data that is both comprehensive and detailed. The primary advantage of this feature fusion strategy lies in its capacity to enhance the classifier's understanding and interpretation of the data. By amalgamating features derived from various aspects of the sign language gestures, the fused feature vector offers a richer and more nuanced representation of the data. This, in turn, facilitates a more accurate and robust classification performance, as the classifier can leverage a broader spectrum of information to discern between different sign language gestures.

### Support vector machine

In this study, the support vector machine (SVM) is used to classify the fused features. The SVM is a supervised learning algorithm that aims to identify the optimal hyperplane for separating the feature space (*Cortes & Vapnik, 1995*). During the training process, the SVM constructs hyperplanes within a high-dimensional space in order to effectively partition the training dataset into distinct classes. In cases in which the feature set is not linearly separable, a kernel function is employed to facilitate the transformation of the data into a new vector space. The present study employed radial basis function (RBF) kernels. The RBF kernel is a kernel that utilizes a radial (exponential) function to smooth a distance measure (*Amari & Wu, 1999*). In contrast to linear kernels, this particular kernel possesses the ability to effectively address the nonlinear nature of the association between class labels and attributes.

## Evaluation metrics

This research utilizes a variety of evaluation metrics to gauge the effectiveness of the proposed methodology. The key definitions for these metrics are as follows:

True positives (TP): These are cases where the model accurately identifies an example as being part of a particular class.

False positives (FP): These cases occur when the model wrongly labels an example as belonging to a certain class when it does not belong to that class.

False negatives (FN): These happen when the model fails to recognize actual examples of a class, incorrectly marking them as not belonging to that class.

True negatives (TN): These are instances where the model correctly identifies an example as not belonging to a specific class.

The overall effectiveness of the model across different categories is measured by the accuracy (Eq. (25)). This is calculated as the ratio of the number of correctly identified examples to the overall number of examples assessed. For a more nuanced evaluation, the precision and recall metrics are used. The precision (Eq. (26)) is the proportion of accurately identified positive examples out of all examples identified as positive for a specific class. The recall (Eq. (27)) measures how many actual positive examples the model correctly identifies. The F1-score (Eq. (28)) is a combined metric, representing the harmonic mean of the precision and recall and offering a balanced view of these two measures. The formulas are given below:

$$\text{Accuracy} = \frac{\text{Total Correct Predictions}}{\text{Total Number of Predictions}}, \tag{25}$$

$$\text{Precision} = \frac{\text{TP}}{\text{TP} + \text{FP}}, \tag{26}$$

$$\text{Recall} = \frac{\text{TP}}{\text{TP} + \text{FN}}, \tag{27}$$

$$\text{F1} - \text{Score} = 2 \times \frac{\text{Precision} \times \text{Recall}}{\text{Precision} + \text{Recall}}. \tag{28}$$

For a comprehensive evaluation, the macro-average of the precision, recall, and F1-score for each class is calculated. This approach gives equal weight to each category's contribution to the model's overall effectiveness, ensuring a balanced assessment across all classes.

## EXPERIMENTAL STUDIES

The experimental studies discussed in this article were carried out on an Ubuntu 18.04 operating system on a computer with an Intel Core i5-8400 processor, 16 GB RAM, and a 12 GB GeForce GTX 1080 Ti GPU. The PyTorch, PyTorch-Lightning, and scikit-learn libraries were preferred in the software infrastructure used.

The ResNet-18 models used in this study are enriched with a dropout layer with a probability of 0.4 just before the fully connected layer to prevent overfitting. During the training of the models, the stochastic gradient descent (SGD) optimization algorithm was

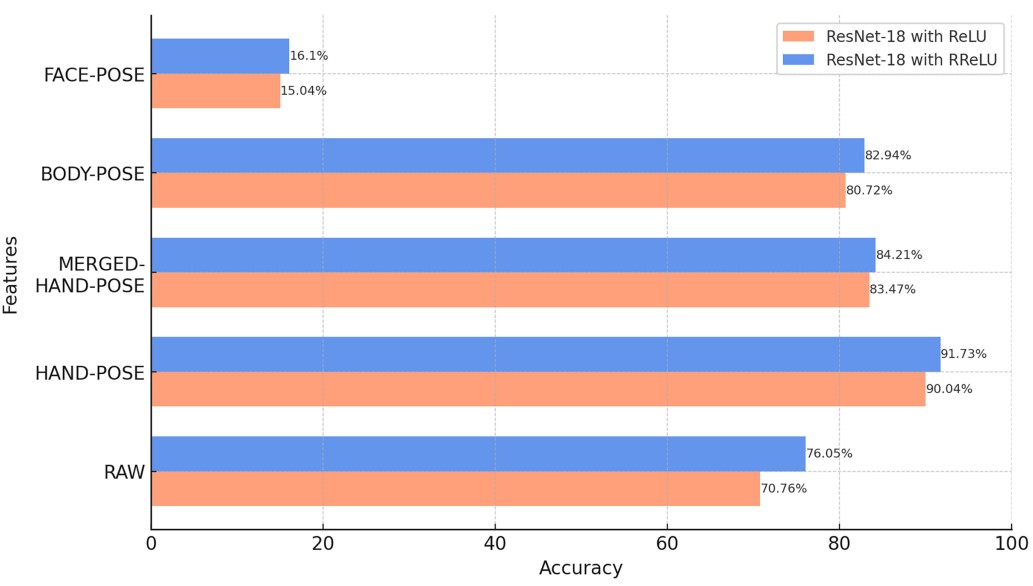

**Figure 7 Results of the classification of raw and pose data with ResNet-18 (BosphorusSign22k-general dataset).**

used, and training was performed for a total of 35 epochs. During the training process, the batch size was set to 8 and the learning rate was initially set to 1e−3 and decreased by 1/10 after every 15 epochs. In addition to the standard ReLU activation function, we also evaluated variations of the models that used the RReLU activation function. The experiments were conducted using the BosphorusSign22k-general dataset, whereas the proposed final method was evaluated on other datasets.

## Classification results of pose images

Initially, the ResNet-18 models were trained using raw video data and four different types of pose data generated from these data. In particular, this experiment compares the performance of the models trained with raw video data with those trained with pose data to compare the accuracy of SLR for each feature. The effect of using the RReLU activation function instead of ReLU is also demonstrated.

According to Fig. 7, the comparison between the model activation functions shows that ResNet-18 models trained with RReLU show a higher classification accuracy than the model trained with ReLU for all five features. These results confirm that the activation function can play a critical role in the performance of deep learning models.

When the classification results obtained are analyzed, HAND-POSE show higher accuracy rates than the other categories. This indicates that hand gestures play a critical role in sign language classification. This apparent success in HAND-POSE shows that the essence of sign language is largely based on gestures made with the hands.

The MERGED-HAND-POSE feature contains more details of the hand shape, orientation, and so on. This detail is achieved by cropping and merging the hand regions from the main image. This process allowed the hand regions, which are normally smaller and indistinct in the overall image, to be visually represented in a larger and more detailed

structure for the MERGED-HAND-POSE feature. However, despite this detailed information, the MERGED-HAND-POSE feature does not include the position information of the hands that is available in the HAND-POSE feature, which is an important deficiency in the classification process. Knowledge of the position of the hands plays a critical role in the classification of isolated sign language words, as some signs are communicated through position changes. Therefore, the loss of this information by the MERGED-HAND-POSE feature leads to a significant decrease in the classification accuracy. These findings show how critical hand position information is in the selection of features to be used in SLR systems.

BODY-POSE features are critical for sign language classification. These features cover the general movements and positions of the body and how other parts of the body interact with hand movements. However, the accuracy of the BODY-POSE feature was slightly lower than that of the hand features (HAND-POSE and MERGED-HAND-POSE). The main reason for this is that hand movements and positions have a more decisive role in determining the meaning of sign language words. Detailed movements of the hands can directly affect the meaning of many sign language words, whereas general body movements have an indirect role in this interpretation. Additionally, since the BODY-POSE features cover a larger area of the body, detailed hand movements are less prominent in this larger area. This could be another factor affecting the classification accuracy of body features.

BODY-POSE features, which are direct pose representations of raw video data, showed a higher classification accuracy than raw video data. This demonstrates the advantage of pose data over raw images. This is because pose data focus only on the motion or gesture and largely eliminate the variations that may arise from the background and the signer's physical features, such as their hair, beard/mustache, clothes, or clothing style. Therefore, these features increase the generalizability of the pose data and make these data superior to raw video data in classification.

The low accuracy of the FACE-POSE classification indicates that facial features alone are not fully sufficient for sign language classification. However, sign language is not limited to hand gestures; facial expressions and body language also play an important role. Therefore, the integrated use of facial features in the model is critical to achieving more accurate classifications.

## Classification results of POSE-MHI features

In this subsection, we evaluate the classification performance of ResNet-18 with different MHI features. The findings presented in Fig. 8 reflect the test accuracies obtained with various MHI features. This investigation is part of our goal to explore how effectively MHI features can capture temporal information in sign language videos, alongside the spatial features evaluated in the previous subsection.

The findings presented in Fig. 8 demonstrate that ResNet-18 models trained using the RReLU activation function exhibit superior accuracy rates for MHI features compared to models trained with the ReLU activation function. This finding corroborates the findings

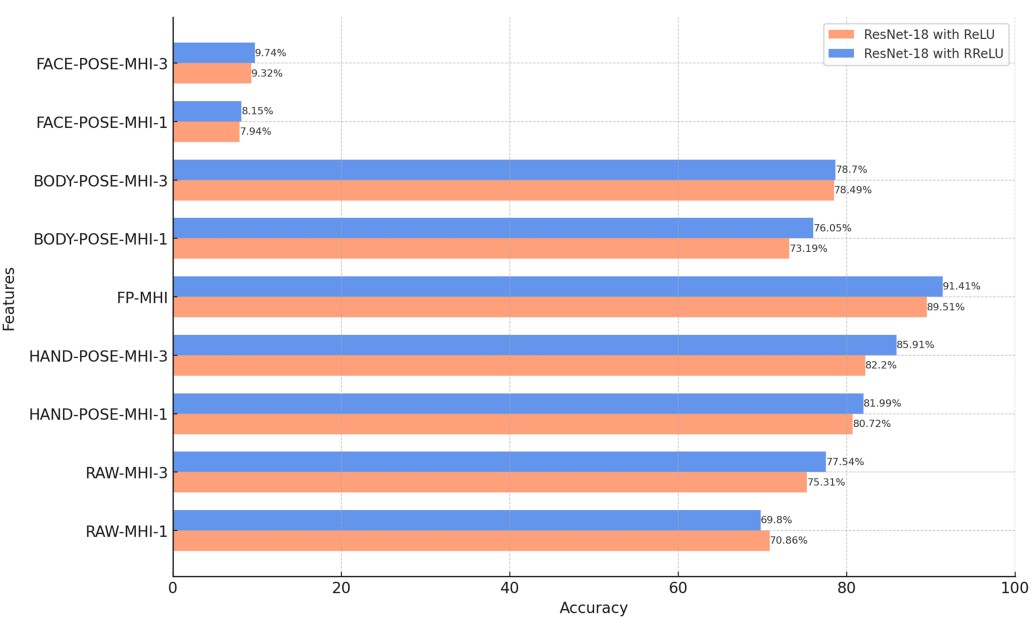

**Figure 8** **Results of the classification of MHI-based features with ResNet-18 (BosphorusSign22k-general dataset).**

observed in the analysis of pose data. It is conceivable that RReLU increases the learning capacity of the model, allowing it to obtain more generalizable and recognizable features.

The results suggest that pose-based MHI features, including BODY-POSE-MHI-1, BODY-POSE-MHI-3, HAND-POSE-MHI-1, HAND-POSE-MHI-3, and FP-MHI, exhibit higher accuracy rates compared to MHI features derived from unprocessed video data, namely RAW-MHI-1 and RAW-MHI-3. The potential benefits associated with the use of pose-based MHI features in the context of SLR are demonstrated by the findings of this study. Furthermore, this research provides the opportunity to perform a comparative analysis between single-channel features and three-channel MHI features. Based on the findings, it can be concluded that the use of three-channel MHI (MHI-3) features yields a more accurate result compared to the implementation of single-channel MHI (MHI-1) features. The MHI-1 technique generates a single representation that covers all motion over the entire video duration, whereas the MHI-3 strategy divides the temporal interval into three different phases. This increases the granularity of the data available for the classification and analysis of sign language.

The proposed feature, known as the FP-MHI, exhibited the highest level of accuracy compared to the other features examined, achieving an impressive rate of 91.41%. This superior performance of the FP-MHI is due to its capacity to provide discriminative information for sign language words. The role of finger positions and movements is of the utmost importance in the communication process of sign language. This suggests that the accurate recognition and classification of finger movements and positions can significantly improve the overall SLR performance.

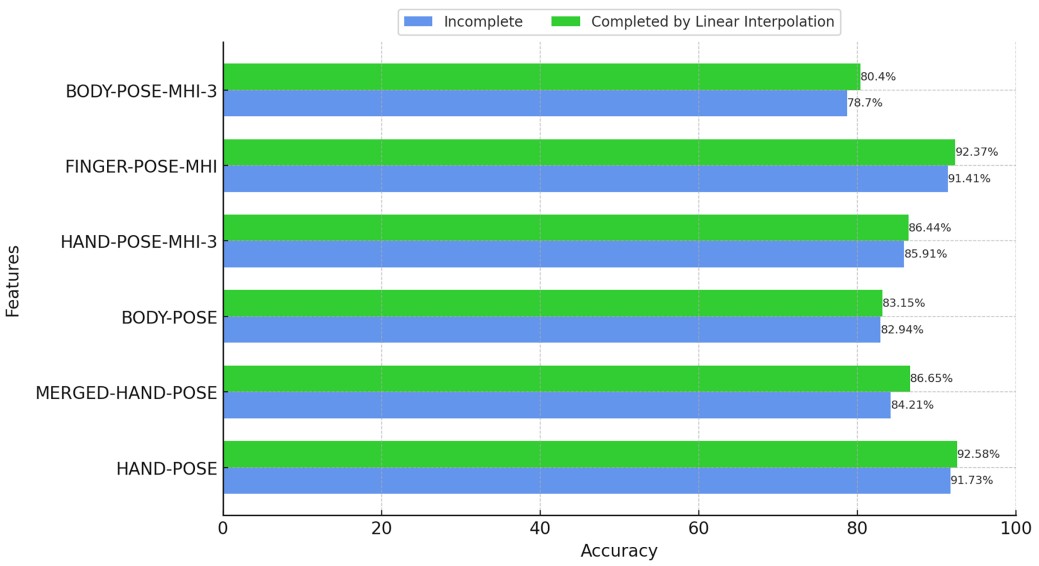

**Figure 9** **Effect of filling in missing poses with linear interpolation (BosphorusSign22k-general dataset).**

In the following subsections of the article, the proposed system is implemented using the ResNet-18 model with the RReLU activation function. Furthermore, MHI-3 features are used, together with the pose data and FP-MHIs.

## Effect of filling in missing poses with linear interpolation

Although the performance of MediaPipe is generally satisfactory, some pose data may not be accurately estimated due to fast hand movements or occlusion. In order to overcome such shortcomings, we chose to complete the missing pose data using linear interpolation when performing feature extraction based on hand poses. The classification results based on the completed features are presented in Fig. 9. In addition, in order to evaluate the effectiveness of linear interpolation, the performance results obtained with incomplete pose data are also presented in the same figure.

According to the results in Fig. 9, it can be seen that the completion of missing pose data using linear interpolation leads to an overall increase in the classification performance. Missing data can cause the model to stumble, especially when it is analyzing dynamic movements and rapid changes. Therefore, filling the gaps using linear interpolation allows the model to classify such motions more accurately. Furthermore, these results reveal how effective even simple and computationally efficient methods such as linear interpolation can be in solving the missing data problem. This is especially important when working with large datasets and complex model structures, as the application of such methods can save both time and resources.

## Fusion and classification results of hand properties

In this subsection, we fuse the features extracted from the flatten layers of the ResNet-18 models trained using HAND-POSE, MERGED-HAND-POSE, HAND-POSE-MHI-3, and FP-MHI in various combinations and classify them with the SVM. The results of these

**Table 4 Fusion of the hand features in different combinations and classification results with the SVM (BosphorusSign22k-general dataset).**

| HAND-POSE | HAND-POSE-MHI-3 | MERGED-HAND-POSE | FP-MHI | Accuracy (in %) | Precision (in %) | Recall (in %) | F1-Score (in %) |
|---|---|---|---|---|---|---|---|
| ✓ | ✓ | | | 92.83 | 92.24 | 93.22 | 91.85 |
| ✓ | | ✓ | | 94.09 | 93.42 | 93.48 | 92.68 |
| ✓ | | | ✓ | 93.78 | 93.10 | 93.96 | 92.65 |
| | ✓ | ✓ | | 93.46 | 92.68 | 93.17 | 91.97 |
| | ✓ | | ✓ | 92.93 | 92.20 | 93.09 | 91.73 |
| | ✓ | ✓ | ✓ | 94.09 | 93.42 | 93.54 | 92.64 |
| ✓ | ✓ | ✓ | | 94.62 | 93.92 | 94.25 | 93.26 |
| ✓ | ✓ | | ✓ | 94.41 | 93.73 | 94.45 | 93.38 |
| ✓ | | ✓ | ✓ | 94.73 | 94.09 | 94.47 | 93.47 |
| | ✓ | ✓ | ✓ | 94.83 | 94.19 | 94.92 | 93.70 |
| ✓ | ✓ | ✓ | ✓ | 95.36 | 94.77 | 95.44 | 94.29 |

operations are presented in Table 4. This approach is based on the hypothesis that combining hand pose data with hand gestures, which are captured through MHIs and encompass both static and dynamic movements, can potentially improve the SLR accuracy by providing a more robust set of features. The need for such a fusion stems from the complex nature of sign language, which often requires a multi-faceted feature set to accurately interpret the rich variety of hand gestures.

Table 4 illustrates the impact of different combinations of hand features on not only the classification accuracy but also the precision, recall, and F1-score. While certain hand features alone demonstrate a high classification accuracy, the fusion of these features plays a crucial role in elevating the overall performance. Notably, the accuracy of 92.83% achieved by combining HAND-POSE and HAND-POSE-MHI-3 is significantly enhanced to 94.62% with the addition of MERGED-HAND-POSE. However, the inclusion of FP-MHIs proves to be decisive in further improving the performance, culminating in a classification accuracy of 95.36%, along with corresponding increases in the precision, recall, and F1-score. This enhancement underscores the importance of integrating FP-MHIs with other hand features, markedly improving the classification accuracy and the overall robustness of the model.

The in-depth examination of hand characteristics is vital in the categorization of sign language. The findings from this research corroborate the idea that hand features, and their various combinations, have a substantial impact on the accuracy, precision, recall, and F1-score of the algorithms employed in sign language classification, emphasizing the importance of a comprehensive feature set for effective interpretation.

## Impact of non-manual features on classification performance

In sign language, not only the position and movements of the hands but also body and facial movements are an important part of communication. In the realm of communication, facial expressions and body language are not only supplementary but play

**Table 5 Fusion of non-manual features with hand features and classification results with the SVM (BosphorusSign22k-general dataset).**

| HAND-FEATURES | BODY-FEATURES | FACE-FEATURES | Accuracy (in %) | Precision (in %) | Recall (in %) | F1-score (in %) |
|---|---|---|---|---|---|---|
| ✓ | | | 95.36 | 94.77 | 95.44 | 94.29 |
| | ✓ | | 88.61 | 87.96 | 90.70 | 87.54 |
| | | ✓ | 14.33 | 14.38 | 18.98 | 13.18 |
| ✓ | ✓ | | 95.89 | 95.30 | 96.30 | 95.07 |
| ✓ | | ✓ | 95.57 | 94.98 | 95.99 | 94.63 |
| | ✓ | ✓ | 86.93 | 86.09 | 90.01 | 85.69 |
| ✓ | ✓ | ✓ | 96.94 | 96.43 | 96.99 | 96.31 |

a pivotal role in conveying meanings and categorizing concepts. Within this framework, our study delves into the integration of hand features with body and facial expressions and examines the impact of these combinations on the accuracy of the classification. The results stemming from the amalgamation of hand features have been previously discussed. Here, we analyze the results of integrating BODY-FEATURES and FACE-FEATURES in different combinations. BODY-FEATURES are obtained by fusing BODY-POSE and BODY-POSE-MHI-3 features, and FACE-FEATURES are obtained by fusing FACE-POSE and FACE-POSE-MHI-3 features.

Table 5 shows the fusion of non-manual features with hand features and their classification results using the SVM in the BosphorusSign22k-general dataset. The table includes the test accuracy, precision, recall, and F1-score for different feature combinations.

The analysis of the table reveals that the classification accuracy for hand features alone is 95.36%, with corresponding precision, recall, and F1-score values of 94.77%, 95.44%, and 94.29%, respectively. When the BODY and FACE features are used independently, their accuracies are 88.61% and 14.33%, with the fusion of the HAND and BODY features increasing the accuracy to 95.89%. This combination achieves a slightly higher accuracy, precision, recall, and F1-score compared to the HAND and FACE feature combination, which results in an accuracy of 95.57%, a precision of 94.98%, a recall of 95.99%, and an F1-score of 94.63%. Using only the BODY and FACE features yields an accuracy of 86.93%, the highest achieved without incorporating HAND features. The highest classification accuracy of 96.94%, along with a precision of 96.43%, recall of 96.99%, and F1-score of 96.31%, is achieved by combining all the HAND, BODY, and FACE features. The fusion of HAND features with BODY and FACE features is crucial for attaining a higher accuracy in sign language classification. However, the contribution of the facial features alone to accuracy is relatively low, suggesting that facial gestures are more effective when they are combined with other features rather than being used independently.

## Comprehensive performance analysis of the proposed method

In our study, we conducted a detailed performance analysis of our proposed SLR method, focusing on the precision, recall, and F1-scores for various signs and examining

**Table 6 Performance analysis of each class in the proposed method (BosphorusSign22k-general dataset).**

| Sign | Prec. | Recall | F1-score | Sign | Prec. | Recall | F1-score | Sign | Prec. | Recall | F1-score |
|---|---|---|---|---|---|---|---|---|---|---|---|
| Explain | 1.00 | 1.00 | 1.00 | Early | 0.40 | 1.00 | 0.50 | Morning | 1.00 | 1.00 | 1.00 |
| Name | 1.00 | 1.00 | 1.00 | Home address | 1.00 | 1.00 | 1.00 | Hide | 1.00 | 1.00 | 1.00 |
| Address | 0.80 | 1.00 | 0.80 | Delay_Late | 1.00 | 1.00 | 1.00 | Sell | 1.00 | 1.00 | 1.00 |
| Relative | 1.00 | 1.00 | 1.00 | Come | 0.80 | 1.00 | 0.80 | Now | 1.00 | 1.00 | 1.00 |
| Evening | 1.00 | 1.00 | 1.00 | To bring | 1.00 | 1.00 | 1.00 | After | 1.00 | 1.00 | 1.00 |
| Dinner | 1.00 | 1.00 | 1.00 | Go | 1.00 | 1.00 | 1.00 | Ask | 1.00 | 1.00 | 1.00 |
| Receive | 1.00 | 1.00 | 1.00 | Dress up | 1.00 | 1.00 | 1.00 | Question | 1.00 | 1.00 | 1.00 |
| Mom | 1.00 | 1.00 | 1.00 | Hide | 1.00 | 1.00 | 1.00 | Surname | 0.80 | 1.00 | 0.80 |
| Call | 1.00 | 1.00 | 1.00 | Send | 1.00 | 1.00 | 1.00 | Say | 1.00 | 1.00 | 1.00 |
| Search (Phone) | 1.00 | 1.00 | 1.00 | See | 1.00 | 1.00 | 1.00 | Undress | 1.00 | 1.00 | 1.00 |
| Friend | 1.00 | 1.00 | 1.00 | Show | 1.00 | 1.00 | 1.00 | Be grateful for | 1.00 | 1.00 | 1.00 |
| Jump | 1.00 | 1.00 | 1.00 | Week | 0.90 | 0.75 | 0.80 | Thirst | 1.00 | 1.00 | 1.00 |
| Moon | 1.00 | 1.00 | 1.00 | Remember | 1.00 | 1.00 | 1.00 | Being Quiet | 1.00 | 1.00 | 1.00 |
| Stand up | 1.00 | 1.00 | 1.00 | Immediately | 1.00 | 1.00 | 1.00 | Know | 1.00 | 1.00 | 1.00 |
| Little | 1.00 | 1.00 | 1.00 | Feel | 1.00 | 1.00 | 1.00 | Repeat | 1.00 | 1.00 | 1.00 |
| Father_Male | 1.00 | 1.00 | 1.00 | Inside | 1.00 | 1.00 | 1.00 | Telephone | 1.00 | 1.00 | 1.00 |
| Yell | 1.00 | 1.00 | 1.00 | Drink | 1.00 | 1.00 | 1.00 | Thank | 1.00 | 1.00 | 1.00 |
| Look | 1.00 | 1.00 | 1.00 | Signature | 1.00 | 1.00 | 1.00 | Forget | 1.00 | 1.00 | 1.00 |
| Like | 1.00 | 1.00 | 1.00 | God willing | 1.00 | 1.00 | 1.00 | Cold | 1.00 | 1.00 | 1.00 |
| Wait | 1.00 | 1.00 | 1.00 | Business address | 0.60 | 0.75 | 0.60 | Exist | 1.00 | 0.71 | 0.80 |
| To look like | 1.00 | 1.00 | 1.00 | Mark | 1.00 | 1.00 | 1.00 | Give | 1.00 | 0.83 | 0.90 |
| To be fed up | 1.00 | 0.83 | 0.90 | To Bite | 1.00 | 1.00 | 1.00 | Catch | 1.00 | 1.00 | 1.00 |
| Know | 0.60 | 1.00 | 0.70 | Heat | 1.00 | 1.00 | 1.00 | Burn | 1.00 | 1.00 | 1.00 |
| A week ago | 0.40 | 0.67 | 0.50 | Hearing | 1.00 | 1.00 | 1.00 | Response | 1.00 | 1.00 | 1.00 |
| You are welcome | 1.00 | 1.00 | 1.00 | Request | 1.00 | 1.00 | 1.00 | Make | 1.00 | 1.00 | 1.00 |
| Some | 1.00 | 0.91 | 0.90 | Good | 1.00 | 0.83 | 0.90 | Year_Age | 1.00 | 1.00 | 1.00 |
| End | 1.00 | 0.83 | 0.90 | Escape | 1.00 | 1.00 | 1.00 | Write | 1.00 | 1.00 | 1.00 |
| Drowning | 1.00 | 1.00 | 1.00 | Woman_Girl | 1.00 | 1.00 | 1.00 | Eat | 1.00 | 1.00 | 1.00 |
| Divorce | 1.00 | 1.00 | 1.00 | Breakfast | 0.80 | 1.00 | 0.80 | Not available | 1.00 | 1.00 | 1.00 |
| Exchange | 1.00 | 1.00 | 1.00 | Close | 0.50 | 1.00 | 0.60 | Swallow | 1.00 | 1.00 | 1.00 |
| Find | 1.00 | 1.00 | 1.00 | Itch | 1.00 | 1.00 | 1.00 | Thinking | 1.00 | 1.00 | 1.00 |
| Invite, call | 0.80 | 1.00 | 0.80 | Fold | 1.00 | 0.71 | 0.80 | Yell_2 | 1.00 | 1.00 | 1.00 |
| Work | 1.00 | 1.00 | 1.00 | Win | 1.00 | 1.00 | 1.00 | Look_2 | 0.80 | 1.00 | 0.80 |
| Stealing (Theft) | 1.00 | 1.00 | 1.00 | Cut | 1.00 | 1.00 | 1.00 | To look like_2 | 1.00 | 0.80 | 0.80 |
| Playing (Music) | 1.00 | 1.00 | 1.00 | Gaining weight | 1.00 | 1.00 | 1.00 | To look like_3 | 0.00 | 0.00 | 0.00 |
| Collide | 1.00 | 1.00 | 1.00 | Lose weight | 1.00 | 1.00 | 1.00 | End_2 | 1.00 | 1.00 | 1.00 |
| Pull | 1.00 | 1.00 | 1.00 | ID | 1.00 | 1.00 | 1.00 | Drowning_2 | 1.00 | 1.00 | 1.00 |
| Pocket | 1.00 | 0.83 | 0.90 | To concentrate | 1.00 | 1.00 | 1.00 | Work_2 | 1.00 | 0.75 | 0.80 |
| Mobile phone | 1.00 | 1.00 | 1.00 | Run | 1.00 | 1.00 | 1.00 | Pull_2 | 1.00 | 1.00 | 1.00 |
| Translate | 1.00 | 1.00 | 1.00 | Bad | 1.00 | 1.00 | 1.00 | Translate_2 | 1.00 | 0.83 | 0.90 |
| Child | 1.00 | 0.62 | 0.70 | Vomit | 1.00 | 1.00 | 1.00 | Very_2 | 0.80 | 1.00 | 0.80 |

(Continued)

| Sign | Prec. | Recall | F1-score | Sign | Prec. | Recall | F1-score | Sign | Prec. | Recall | F1-score |
|---|---|---|---|---|---|---|---|---|---|---|---|
| Children | 0.60 | 1.00 | 0.70 | God Preserve From Evil | 1.00 | 1.00 | 1.00 | Listen_2 | 1.00 | 1.00 | 1.00 |
| Very | 1.00 | 1.00 | 1.00 | How are you | 1.00 | 1.00 | 1.00 | Touch_2 | 0.80 | 0.67 | 0.70 |
| Collapse | 1.00 | 1.00 | 1.00 | Number | 1.00 | 1.00 | 1.00 | Touch_3 | 1.00 | 1.00 | 1.00 |
| State | 0.80 | 1.00 | 0.80 | Pay | 1.00 | 1.00 | 1.00 | Beating_2 | 1.00 | 1.00 | 1.00 |
| Listen | 1.00 | 1.00 | 1.00 | Boy_Male | 1.00 | 1.00 | 1.00 | Email_2 | 1.00 | 1.00 | 1.00 |
| Outside | 1.00 | 1.00 | 1.00 | Noon | 1.00 | 1.00 | 1.00 | Breastfeeding_2 | 1.00 | 1.00 | 1.00 |
| Pour | 1.00 | 1.00 | 1.00 | Lunch | 1.00 | 1.00 | 1.00 | Send_2 | 1.00 | 1.00 | 1.00 |
| Touch | 1.00 | 1.00 | 1.00 | Learn | 1.00 | 1.00 | 1.00 | God willing_2 | 1.00 | 1.00 | 1.00 |
| Freeze | 1.00 | 1.00 | 1.00 | Teaching | 1.00 | 1.00 | 1.00 | To Bite_2 | 1.00 | 1.00 | 1.00 |
| Beating | 1.00 | 1.00 | 1.00 | School address | 1.00 | 1.00 | 1.00 | Breakfast_2 | 1.00 | 1.00 | 1.00 |
| Fight | 1.00 | 1.00 | 1.00 | Be | 1.00 | 1.00 | 1.00 | Breakfast_3 | 1.00 | 1.00 | 1.00 |
| Stop | 1.00 | 1.00 | 1.00 | To Die | 1.00 | 1.00 | 1.00 | Breakfast_4 | 1.00 | 1.00 | 1.00 |
| Fall | 1.00 | 1.00 | 1.00 | Before | 1.00 | 1.00 | 1.00 | Lose weight_2 | 0.80 | 1.00 | 0.80 |
| Email | 1.00 | 1.00 | 1.00 | Cover | 1.00 | 0.71 | 0.80 | Number_2 | 1.00 | 1.00 | 1.00 |
| To crawl | 1.00 | 1.00 | 1.00 | Sit | 1.00 | 1.00 | 1.00 | Sell_2 | 1.00 | 0.71 | 0.80 |
| Be retired | 1.00 | 1.00 | 1.00 | Special | 1.00 | 1.00 | 1.00 | After_2 | 0.80 | 1.00 | 0.80 |
| Breastfeeding | 1.00 | 1.00 | 1.00 | Hour (Time) | 1.00 | 1.00 | 1.00 | Make_2 | 1.00 | 1.00 | 1.00 |

**Table 7 Misclassification signs (BosphorusSign22k-general dataset).**

| Sign | Confused sign | Num. of confused signs | Sign | Confused sign | Num. of confused signs | Sign | Confused sign | Num. of confused signs |
|---|---|---|---|---|---|---|---|---|
| Address | Business Address | 1 | Early | Touch_2 | 2 | Surname | Translate_2 | 1 |
| Know | Exist | 2 | Come | Pocket | 1 | Look_2 | Some | 1 |
| A week ago | Week | 3 | Week | A week ago | 1 | To look like_3 | To look like_2 | 2 |
| Invite, call | Sell_2 | 1 | Business Address | Work_2 | 2 | Very_2 | Good | 1 |
| Children | Child | 2 | Breakfast | Cover | 1 | Touch_2 | To be fed up | 1 |
| State | Give | 1 | Close | Cover | 1 | Lose weight_2 | Child | 1 |
| Early | End | 1 | Close | Fold | 2 | After_2 | Sell_2 | 1 |

misclassification examples. This comprehensive evaluation is important to understand the effectiveness of the model and areas for improvement.

Table 6 shows the high performance of our model in correctly identifying most sign language words. This highlights the model's strong capability to accurately recognize a diverse range of signs, showcasing its robustness and reliability. However, certain words, such as "early," "address," and "come," exhibit a lower precision, underscoring specific challenges within the model. These may stem from the need for more varied training data or improvements in the feature extraction processes. In addition, the method generally

retains high recall scores for signs with low precision. This suggests that although the model may confuse certain signs with others, it is still able to reliably detect their presence.

Table 7 provides insights into the model's behavior and limitations through its detailed confusion analysis. It shows how the model confuses similar signs, such as "address" and "business address" or "children" and "child," indicating challenges in distinguishing subtle differences. This could be due to similarities in the gesture dynamics, hand shapes, or movement patterns inherent to these sign pairs. These findings emphasize the importance of enriching the training dataset with a wider range of examples, in particular focusing on physically similar signs.

In summary, our analysis comprehensively showcases the strong aspects of our model for SLR. This study not only emphasizes the capabilities of our current model but also provides valuable contributions to the ongoing research in the field of SLR and interpretation. The development of training strategies and feature extraction techniques that will further enhance the model's performance represents significant steps in advancing this field.

## Temporal and cost analysis of the proposed method

In this subsection, we perform a temporal and cost analysis of our proposed SLR system. These analyses are performed on hardware with an Intel Core i5-8400 processor, 16 GB RAM, and a GeForce GTX 1080 Ti GPU, which was also used in the design of the system. We first compare the ResNet-18 model used in the proposed system with other efficient state-of-the-art models, and then we evaluate the inference time performance of each component used in our proposed final SLR system. We also perform some optimizations to improve the temporal performance and examine the inference time using the entire test set.

We first compare the performance of the ResNet-18 model used in our proposed SLR system with a set of state-of-the-art models with parameters equal to or less than those of ResNet-18. These models are MobileNetV2 (*Sandler et al., 2018*), MobileNetV3 (*Howard et al., 2019*), EfficientNet-(B0, B1, B2, B3) (*Tan & Le, 2019*), SqueezeNet1-0 (*Iandola et al., 2016*), and Densenet121 (*Huang, Liu & Weinberger, 2016*). For this purpose, we trained all these models using FP-MHI data for 35 epochs with the SGD optimizer in the same way that we trained ResNet-18, setting the initial learning rate to 1e−3 and reducing it by 1/10 every 15 epochs. Table 8 contains a comparison of these trained efficient models and ResNet-18. This comparison is based on several metrics: the test accuracy, parameter size, training time on GPU, inference time on GPU and CPU, total memory usage, and million floating-point operations per second (MFLOPS) value.

According to the comparison results in Table 8, the ResNet-18 model with the RReLU activation function outperformed the other models, especially in terms of the test accuracy, with a success rate of 92.37%. This performance is one of the reasons that ResNet-18 is the preferred model in our proposed deep learning-based SLR systems. The training time is relatively low for ResNet-18, with 10.75 s per epoch on GPU, while SqueezeNet has a faster training time. However, the accuracy of ResNet-18 outperformed the model with these faster training times. ResNet-18 offers very competitive inference times compared to other

**Table 8 Comparison of ResNet-18, used in the proposed system, with other efficient models.**

| Models | Test accuracy (%) | Total params | Training time on GPU per epoch (second) | Inference time on GPU (millisecond) | Inference time on CPU (millisecond) | Total memory usage (MB) | MFLOPS |
|---|---|---|---|---|---|---|---|
| MobileNetV2 | 90.57 | 2.4 M | 13.99 | 3.48 | 6.18 | 37.56 | 91.34 |
| MobileNetV3 | 83.36 | 5.7 M | 15.63 | 4.28 | 6.70 | 41.20 | 68.33 |
| EfficientNet-B0 | 89.40 | 4.2 M | 19.35 | 5.53 | 8.94 | 45.24 | 117.42 |
| EfficientNet-B1 | 88.34 | 6.7 M | 26.03 | 7.76 | 13.73 | 66.11 | 173.87 |
| EfficientNet-B2 | 89.30 | 7.9 M | 26.56 | 7.95 | 13.27 | 72.87 | 199.97 |
| EfficientNet-B3 | 89.40 | 11.0 M | 31.46 | 8.85 | 17.48 | 98.06 | 287.39 |
| SqueezeNet1-0 | 84.21 | 824 K | 8.58 | 1.52 | 3.88 | 11.68 | 174.84 |
| Densenet121 | 91.63 | 7.1 M | 30.69 | 9.68 | 18.98 | 73.76 | 722.58 |
| ResNet-18 | 92.37 | 11.3 M | 10.75 | 1.82 | 6.95 | 55.71 | 489.79 |

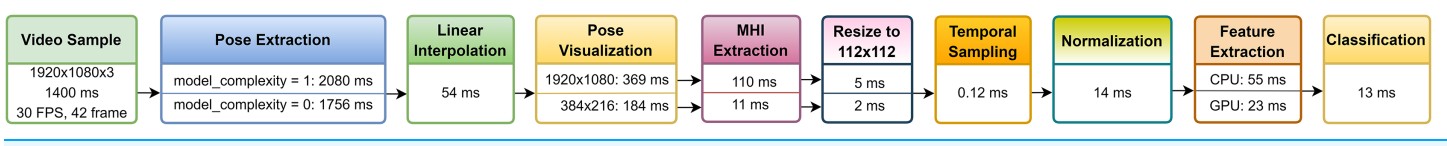

**Figure 10 Temporal performance analysis of each process in the proposed SLR system.**

models, with values of 1.82 ms on GPU and 6.95 ms on CPU. This provides a significant advantage for real-time applications. The total memory usage was also reasonable at 55.71 MB, indicating that ResNet-18 can run effectively even on resource-constrained systems. The models were also evaluated in terms of MFLOPS values. The MFLOPS value refers to the maximum number of floating-point operations a model can perform in one second and is often used to evaluate the computational intensity and efficiency of a model. ResNet-18's MFLOPS value of 489.79 indicates that the model requires relatively modest computational resources while providing a high accuracy, making it suitable for resource-constrained devices. This analysis reveals that ResNet-18 is a good choice for our proposed SLR task in terms of both accuracy and performance. Hence, the use of ResNet-18 in our proposed SLR system plays a critical role in improving the overall performance and effectiveness of our system.

Following the evaluation of the ResNet-18 model, a comprehensive analysis of the processing times of each process in our proposed final SLR system, which we created using eight ResNet-18s, was performed on a sample video taken from the BosphorusSign22k-general dataset consisting of 42 frames (1,920 × 1,080 resolution, 1,400 ms, 30 FPS). For this sample video, the system was run 100 times and average values were obtained. In addition, some optimizations were made to improve the temporal performance of the system. These optimizations included changing the model_complexity parameter of MediaPipe, rendering pose visualizations at a lower resolution, and performing feature

extraction on the GPU. The temporal performance of each process is shown in milliseconds in the diagram in Fig. 10.

The total processing time was measured as 2,700 ms, including all operations starting from the extraction of the pose data and ending with the SVM classification. This time reflects the integrated operation of all components of the system. Pose extraction took 2,080 ms, the linear interpolation of missing poses took 54 ms, pose visualization took 369 ms, the calculation of MHI features took 110 ms, temporal sampling to select 32 frames took 0.12 ms, normalization took 14 ms, feature extraction from the ResNet-18 models took 55 ms, and SVM classification took 13 ms.

Since pose data are extracted from each frame in the video separately, pose data can be extracted during the reading of the video from a source or during the display of the video. Therefore, in order to extract pose data from a 1,400 ms video, it is expected that the video has been read from start to finish. This means that we can ignore the video duration for this exposure extraction time. Thus, for a video of roughly 1,400 ms, we can say that the pose extraction time cost is 680 ms. To confirm this, we showed the sample video on screen from start to finish and extracted the pose data. We saw that the video playback took 2,142 ms at about 20 FPS. By the end of the video playback, all pose data had been extracted. Considering the cost of displaying the video on the screen, we can say that pose data were obtained with a delay of 742 ms.

Pose data extraction accounts for approximately 77% of the total time of our system. This process is the most time-consuming step, and any improvement in this step will increase the temporal performance of the system. For this purpose, we updated the default value of MediaPipe's model_complexity parameter from 1 to 0 and performed inference through a less complex model. As a result, the pose extraction time decreased from 2,080 to 1,756 ms, a gain of 324 ms. Another optimization that we made is in the pose visualization step. By reducing the 1,920 × 1,080 resolution pose images we created in this step by one-fifth, i.e., to 384 × 216, we saved 185 ms in the pose visualization process and created the relevant images in 145 ms. These images, which we created with a lower resolution, also affected the MHI extraction and "resize to 112 × 112" processes in the next processing steps, resulting in a total gain of 288 ms. We also performed feature extraction from the trained ResNet-18 models on the CPU by default. When we performed this processing step on the GPU, we achieved a gain of 22 ms. With these optimizations, our proposed method achieved a gain of 643 ms and an extraction time of 2,057 ms for a 1,400 ms sample video. When we ignore the video duration, we can say that the necessary operations were completed in 657 ms. These optimizations are an important step for our system to be used more effectively in real-time applications.

Another important aspect of these optimizations is the impact on memory usage. In our standard solution, the system used about 1.7 GB of memory, whereas after the optimizations, the memory usage was reduced to about 1.2 GB. This is a reduction of about 30% and allows the system to run more efficiently, especially on resource-constrained devices. Less memory usage also allows our system to be integrated in more applications, enabling more efficient multitasking.

**Table 9 Inference times and accuracies with temporal optimizations on the BosphorusSign22k-general test set.**

| MediaPipe model_complexity | Pose visualization resolution | Time (seconds) | Accuracy (in %) |
|---|---|---|---|
| 1 | 1,920 × 1,080 | 4,965 | 96.94 |
| 0 | 1,920 × 1,080 | 4,344 | 96.73 |
| 1 | 384 × 216 | 4,357 | 94.62 |
| 0 | 384 × 216 | 3,739 | 94.52 |

**Table 10 Comparison with results in the literature (BosphorusSign22k-general dataset).**

| Studies | Input modality | Params | Methodology | Accuracy (in %) |
|---|---|---|---|---|
| *Kindiroglu, Ozdemir & Akarun (2019)* | Pose | Full frame | TAF & hue subunit detection, CNN | 81.58 |
| *Gündüz & Polat (2021)* | RGB, pose, optical flow | Body, hand, face | Inception 3D & LSTM-RNN fusion | 89.35 |
| Proposed method (FP-MHI) | MHI | Finger | Resnet-18 with RReLU | 92.37 |
| Proposed method (Fusing + SVM) | Pose, MHI | Body, hand, face, finger | Resnet-18 with RReLU, Feature Fusion, SVM | 96.94 |

We also evaluated the proposed system, with optimizations for temporal efficiency, using the entire BosphorusSign22k-general test set of 949 videos. This gave us the total inference time, as well as the effect of temporal optimizations on the accuracy. Table 9 shows the inference times and accuracies obtained from the analysis on the test set, consisting of 2,623 s in total.

In Table 9, the performance of the proposed SLR system is evaluated using different MediaPipe model complexities and pose visualization resolutions. The results show that reducing the model complexity (from 1 to 0) reduces the total inference time by about 12.5%, while reducing the resolution reduces the time by about 12.2%. The impact of these optimizations on the accuracy is also indicated in the table. Reducing the model complexity, possibly because it misses some pose data, and reducing the pose visualization resolution, because some details are lost, resulted in decreases in accuracy of 0.21% and 2.32%, respectively. These represent an important trade-off between the system performance and accuracy. These results show that our optimizations for temporal efficiency yield significant improvements, with acceptable reductions in the accuracy.

## Comparison with other studies

After obtaining the results of our experimental work, it is important to compare the findings with other studies in the literature. In this subsection, we compare the performance of our proposed method with that of other studies. The comparisons will include the results obtained with our proposed FP-MHI feature and the final method obtained by fusing all features.

Table 10 presents the comparison of our experimental work on the BosphorusSign22k-general dataset with similar studies in the literature. Our proposed FP-MHI feature stands

**Table 11 Comparison with results in the literature (BosphorusSign22k dataset).**

| Studies | Modality | Params | Methodology | Accuracy (in %) |
|---|---|---|---|---|
| *Özdemir et al. (2020)* | RGB | Full frame | MC3-18 | 78.85 |
| *Özdemir et al. (2020)* | RGB | Full frame | IDT | 88.53 |
| *Gökçe et al. (2020)* | RGB | Body, hand, face | MC3-18, score-level fusion | 94.94 |
| *Sincan & Keles (2021)* | RGB, MHI | Full frame | ResNet-50, I3D | 94.83 |
| *Kındıroglu, Özdemir & Akarun (2023)* | RGB, pose | Full frame | ATAF, TTN, MC3-18, Fusion | 94.90 |
| *Özdemir, Baytaş & Akarun (2023)* | RGB, pose | Body, hand, face | ST-GCN, MC-LSTM | 92.58 |
| Proposed method (FP-MHI) | MHI | Finger | Resnet-18 with RReLU | 86.10 |
| Proposed method (Fusing + SVM) | Pose, MHI | Body, hand, face, finger | Resnet-18 with RReLU, Feature Fusion, SVM | 94.87 |

out with an accuracy of 92.37% compared to the methods of *Kindiroglu, Ozdemir & Akarun (2019)* and *Gündüz & Polat (2021)*. In particular, compared to the complex system based on models such as Inception 3D and LSTM-RNN using body, hand, face, optical flow, and pose data presented by *Gündüz & Polat (2021)*, our relatively simpler ResNet-18 model, in which FP-MHI features are integrated, provided an accuracy increase of 4.02%. This result against a model like Inception-3D that evaluates both spatial and temporal features shows that even with simple models, a high performance can be achieved through proper feature engineering and efficient algorithm design. Furthermore, the integration of body, hand, and face features in our system contributed to a significant test accuracy of 96.94% on the BosphorusSign22k-general dataset. This shows that our proposed method provides a significant improvement over other methods in the existing literature on sign language recognition and can provide a new reference point for research in this area.

Table 11 contains a comparison of methods applied to the BosphorusSign22k dataset. This dataset has rich content for sign language recognition research and covers 744 different classes. The repetition of our high results on the 174-class BosphorusSign22k-general dataset on this dataset shows the effectiveness and scalability of our method. Looking at the results, the approach presented by *Özdemir et al. (2020)*, which extracts both temporal and spatial features from raw video data using the MC3-18 model, shows that our FP-MHI feature provides a 7.25% higher test accuracy compared to the 86.10% accuracy obtained with the ResNet-18 model, according to the IDT feature used in the same study (*Özdemir et al., 2020*). These results once again emphasize the importance of correct feature selection, as well as the complexity of deep learning models, in sign language recognition. *Gökçe et al. (2020)* obtained the highest accuracy rate on the BosphorusSign22k dataset. This work involves the use of the MC3-18 model, a powerful pre-trained model that can handle temporal and spatial information, as well as the integration of various components such as the body, hands, and face. Following a similar approach to these studies, our proposed integrated method achieves a competitive result of 94.87% accuracy with ResNet-18, a simpler model than MC3-18, by combining body, hand, face, and finger features. This result shows that temporal features can be significantly captured by the three-channel MHI features extracted from each piece of pose data.

**Table 12 Comparison with results in the literature (LSA64 dataset).**

| # | Studies | Modality | Params | Methodology | Accuracy (in %) |
|---|---------|----------|--------|-------------|-----------------|
| E1 | *Marais et al. (2022)* | RGB | Full frame | InceptionV3-GRU | 74.22 |
| | Proposed method (FP-MHI) | MHI | Finger | Resnet-18 with RReLU | 94.68 |
| | Proposed method (Fusing + SVM) | Pose, MHI | Body, hand, face, finger | Resnet-18 with RReLU, Feature Fusion, SVM | 98.43 |
| E2 | *Ronchetti et al. (2016)* | RGB | Hand | BoW+SubCls | 91.70 |
| | *Rodríguez & Martínez (2018)* | RGB | Full frame | Cumulative SD-VLAD with SVM | 85.00 |
| | *Alyami, Luqman & Hammoudeh (2023)* | Pose | Hand, face | Transformer | 91.09 |
| | Proposed method (FP-MHI) | MHI | Finger | Resnet-18 with RReLU | 94.99 |
| | Proposed method (Fusing + SVM) | Pose, MHI | Body, hand, face, finger | Resnet-18 with RReLU, Feature Fusion, SVM | 98.68 |
| E3 | *Ronchetti et al. (2016)* | RGB | Hand | BoW+SubCls | 97.00 |
| | *Konstantinidis, Dimitropoulos & Daras (2018b)* | Pose | Body, hands | LSTM | 98.09 |
| | *Konstantinidis, Dimitropoulos & Daras (2018a)* | RGB, pose, optical flow | Body, hand, face | VGG-16, LSTM | 99.84 |
| | *Zhang & Li (2019)* | RGB | Full frame | MEMP network | 99.063 |
| | *Imran & Raman (2020)* | MHI, dynamic image, RGBMI | Full frame | CNN, kernel-based extreme learning machine | 97.81 |
| | *Marais et al. (2022)* | RGB | Full frame | Pruned VGG | 95.50 |
| | *Bohacek & Hruz (2022)* | Pose | Body, hand | Transformer | 100 |
| | *Alyami, Luqman & Hammoudeh (2023)* | Pose | Hand, face | Transformer | 98.25 |
| | Proposed method (FP-MHI) | MHI | Finger | Resnet-18 with RReLU | 98.90 |
| | Proposed method (Fusing + SVM) | Pose, MHI | Body, hand, face, finger | Resnet-18 with RReLU, Feature Fusion, SVM | 100 |

Furthermore, the use of RGB data to train the models in the study by *Gökçe et al. (2020)* and the fact that the dataset was built on a flat background suggests that the system may be potentially sensitive to different background conditions. This could be an important limitation in real-world applications, given the diversity of backgrounds. Our proposed method is based entirely on pose data, which is expected to make our system more robust to background variations. In this context, the success of our pose estimation method is a determining factor for the overall performance of our system. When effective pose estimation is achieved, the sign language recognition system can accurately recognize and interpret sign language components regardless of background variations. This feature is critical for our proposed system to work reliably in various application scenarios.

Table 12 shows the results of the experiments performed on the LSA64 dataset of 64 classes to verify the applicability of our proposed method in different sign languages. This table focuses on the results of the experiments in which the dataset is divided into three different training and testing conditions. E1 and E2 show the signer-independent evaluations, while E3 shows the results when the training and test set are randomly determined. In E1, there is only one previous study. In this evaluation, where signers 5 and

**Table 13 Comparison with results in the literature (GSL dataset).**

| Studies | Modality | Params | Methodology | Accuracy (in %) |
|---|---|---|---|---|
| *Adaloglou et al. (2020)* | RGB | Full frame | GoogLeNet, TConvs | 86.03 |
| *Adaloglou et al. (2020)* | RGB | Full frame | 3D-ResNet, BLSTM | 86.23 |
| *Adaloglou et al. (2020)* | RGB | Full frame | I3D, BLSTM | 89.74 |
| *Selvaraj et al. (2021)* | Pose | Full frame | LSTM | 86.60 |
| *Selvaraj et al. (2021)* | Pose | Full frame | Transformer | 89.50 |
| *Selvaraj et al. (2021)* | Pose | Full frame | ST-GCN | 93.50 |
| *Selvaraj et al. (2021)* | Pose | Full frame | SL-GCN | 95.40 |
| *Fang et al. (2023)* | RGB, pose | Full frame | Adversarial multi-task learning | 91.49 |
| Proposed method (FP-MHI) | MHI | Finger | Resnet-18 with RReLU | 88.55 |
| Proposed method (Fusing + SVM) | Pose, MHI | Body, hand, face, finger | Resnet-18 with RReLU, Feature Fusion, SVM | 95.14 |

10 out of 10 signers are separated as test data, the accuracy rate of 74.22% obtained by *Marais et al. (2022)* is increased to 94.68% with our proposed FP-MHI method and 98.43% with the Fusing+SVM method. These results may be a new starting point for signer-independent evaluations. In E2, where each of the 10 signers was used as a separate test, the experiments were repeated 10 times, and the average accuracies were obtained; the results obtained with our proposed FP-MHI feature and the Fusing+SVM method surpassed all other studies. *Alyami, Luqman & Hammoudeh (2023)*'s work on the classification of keypoints from the signer's hands and face with the transformer model, which was one of the latest studies included in this analysis, falls short of the result obtained by classifying FP-MHIs, which are finger-based features, with ResNet-18. For E3, the 100% accuracy rate obtained by *Bohacek & Hruz (2022)* is equivalent to our proposed methods. In this study, pose data are used in a similar way to our study. This proves the effectiveness of pose data in sign language recognition. For E3, our proposed methodology achieves a 98.90% accuracy with FP-MHIs and a 100% accuracy with Fusing+SVM. Overall, this detailed comparison shows that our proposed methodologies perform competitively and even outperform existing work in the field of sign language recognition, in both the signer-independent and randomized split test conditions.

Another dataset that we used to verify the applicability of our proposed method to different sign languages is the GSL dataset, which contains 310 different Greek Sign Language words. Table 9 shows the studies on the GSL dataset. The same author names refer to the results of different methods from the same article. In the experiments performed on the GSL dataset, the proposed FP-MHI method provides a similar accuracy, especially compared to some recent studies. In particular, it provides results close to the results of *Adaloglou et al. (2020)*. However, our proposed Fusion + SVM method stands out with a higher accuracy rate than almost all studies in the literature. In particular, it is very close to the highest accuracy rate obtained by *Selvaraj et al. (2021)*. In conclusion, the experiments performed on the GSL dataset show that our proposed methods are effective and competitive with other approaches in the literature.

In summary, when the data from Tables 10–13 are analyzed, we can see that our proposed methods provide highly competitive and, in some cases, superior results compared to existing methods in the literature on a wide range of different datasets. In particular, the FP-MHI feature stands out with its performance on various datasets. On some datasets, this feature made it possible to obtain similar or even better results than existing methods in the literature. This impressive performance of the FP-MHI feature shows how valuable and effective this feature is in sign language classification. The achieved accuracy rates of our method, which incorporates the fusion of all features and utilizes an SVM for classification, are comparable to or surpass the highest reported outcomes in the existing literature. This demonstrates that the feature extraction and fusion approach employed in this study is highly efficacious in addressing the sign language classification problem. Moreover, the demonstrated consistency of the experimental results indicates that the employed methodologies possess reliability and robustness across diverse datasets.

## CONCLUSIONS

This study presents an innovative approach for recognizing isolated sign language videos. This approach takes into account both spatial and temporal information by combining pose data from videos and MHI data derived from these pose data. By using improved ResNet-18 models with the RReLU activation function for feature extraction and fusing these features for classification with an SVM, significant improvements in sign language recognition accuracy are achieved. The effectiveness of this approach is demonstrated by competitive or even superior results on the BosphorusSign22k-general, BosphorusSign22k, LSA64, and GSL datasets. In particular, the FP-MHI proposed in this article has made significant progress in this area by emphasizing the fine details of finger gestures, which are often overlooked in sign language recognition studies.

This research has provided clear and concrete answers to our research questions, revealing that the fusion of pose and MHI data has a significant impact on the accuracy of SLR classification. Furthermore, it has been determined that the FP-MHI achieves significant success in terms of the SLR accuracy, and the interaction of this feature with other manual and non-manual features improves the overall performance of SLR systems. It has also been observed that completing missing pose data through linear interpolation positively affects the performance of the models. In addition, the results of the time and cost analysis constitute an important step for the more effective use of our system in real-time applications. This analysis has provided important information about the improvements that can be made to improve the performance of our system. Moreover, since the system is based entirely on pose data, our method is robust to a large degree of background change, depending on the success of the pose data extraction.

In conclusion, we believe that the findings of this study have the potential to make a valuable contribution to the emerging field of SLR research. Our future research in this area will focus on further enhancing the robustness and versatility of our proposed approach by exploring its integration with different classification methodologies, further optimizing the system for real-time applications, and exploring the integration of

additional modalities to improve the recognition of more complex sign language structures. Furthermore, since finger positions and movements are key to capturing the subtle nuances of sign language, focusing more on these data has the potential to further increase the accuracy and sensitivity of our system. Therefore, by paying greater attention to finger data, we will continue our research towards developing advanced SLR systems that can recognize sign language more accurately and effectively.

### Funding
The authors received no funding for this work.

### Competing Interests
The authors declare that they have no competing interests.

### Author Contributions
- Ali Akdağ conceived and designed the experiments, performed the experiments, analyzed the data, performed the computation work, prepared figures and/or tables, and approved the final draft.
- Ömer Kaan Baykan conceived and designed the experiments, performed the experiments, analyzed the data, performed the computation work, authored or reviewed drafts of the article, and approved the final draft.

### Data Availability
The code is available in the Supplemental File.

The datasets are available at:

- BosphorusSign22k is a Turkish Sign Language dataset developed by researchers at Bogazici University (*Camgoz et al., 2016*; *Özdemir et al., 2020*) and is publicly accessible for academic research, provided that an End User License Agreement (EULA) is submitted by the users. Detailed contact information and the process of accessing the dataset can be found on the official dataset website: https://ogulcanozdemir.github.io/bosphorussign22k/.

- LSA-64 (*Ronchetti et al., 2016*) is the Argentine Sign Language dataset provided by III-LIDI (Informatics Institute) at the Informatics Faculty of the National University of La Plata (UNLP) available at: https://facundoq.github.io/datasets/lsa64/.

- GSL (*Adaloglou et al., 2020*) is the Greek Sign Language dataset maintained by the Visual Computing Lab at the Information Technologies Institute available at: https://vcl.iti.gr/dataset/gsl/.

### Supplemental Information
Supplemental information for this article can be found online at http://dx.doi.org/10.7717/peerj-cs.2054#supplemental-information.

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
