# Peer review of "Isolated sign language recognition through integrating pose data and motion history images"

_PeerJ Computer Science, doi:10.7717/peerj-cs.2054_

## Round 0.1 · original submission · Major Revisions

The authors should prepare a major revision based on all comments of the two reviewers.

**Language Note:** The review process has identified that the English language must be improved. PeerJ can provide language editing services - please contact us at [email protected] for pricing (be sure to provide your manuscript number and title). Alternatively, you should make your own arrangements to improve the language quality and provide details in your response letter. – PeerJ Staff

·

Basic reporting

The authors need to become more professional in reporting the data in Tabular format. Just sharing the name with citation and accuracy is little in-sufficient for the basic reporting.
Add a little bit of details in your Tabular accuracy.
I suggest to have a look of reporting in this article:

https://doi.org/10.1016/j.suscom.2023.100907

Also, in the Introduction end, you should have to discuss the organization of the article. You need to be consistent in the Introduction.
The article suddenly changes to topic of Deep Learning without any pre-requisites information for the reader.

Equations needs to be aligned

Experimental design

The investigation is presented in a goof visual format. However it lacks some basic block diagram for understanding the process.

The MHI signals with pose needs to be explained in more intuitive manner. However the visualizations are good.

The ResNet-18 is computationally expensive, Have you compare your proposed methods with earlier methods in terms of time and cost ?

Does your solution have the practical viability to scale up ?

The solution you have discussed about SIFT can be viable to be deployed but what about yours?

You're using a sophisticated background. What are the cases in case of clutter/occlusion/noisy background ?

Validity of the findings

Literature is clearly presented but the author needs to be little consistent. If you are reporting accuracy in the table. Make sure to highlight the gaps when writing the literature survey.

Lacking in a logical context makes it a little boring for the user. Read your Paragraph Survey again and again. You'll find a little shift from reporting methods to talking about accuracy.

However, your method is explained well but you are severely lacking in Mathematical context and equations.

Can we distribute the methods in form of chart : ML Based, DL Based & Traditional Feature Extraction Based methods

Additional comments

Mathematical Equations needs to be added

Dataset Reporting in Intuitive Way

Why you're superior? needs to be discussed

Clear the research questions and give answers in the paper.

Add the little bold section comparing your methods with other popular known ones.

Reviewer 2 ·

Basic reporting

The authors proposed a novel approach for the recognition of isolated sign language videos using Resnet and Support Vector Machines. The paper is interesting but needs major improvements.
First, the abstract does not highlight the actual contribution as SVM and Resnet are well known algorothms. I believe that utlizing these approahces in an effective manner leads to an important contribution, but abstact fails to show the actual working of proposed work.
Second, why authors didnt explored Hourglass network that are specifically designed for post estimation. Some other good allternatives are also missing in discussion such as Densenet.
The title is too confusing, i will suggest do not use abbrevaution and also avoid to lengthen it.
The proposed framewrok is missing leads to a question regarding the major contribution. There is a need of in-depth figure with detail discussion on proposed framework that should highlight the actual contribution.

Experimental design

Included in basic reporting

Validity of the findings

Finding seems valid but needs a detailed comparsion with recent studies.

---

## Round 0.2 · accepted · Accept

The authors have addressed all of the reviewers' comments as submitted by the reviewers.

·

Basic reporting

All things are revised as per my comments

Experimental design

The design is improved as per the suggestions

Validity of the findings

The findings are visually presented as per requested in my comments

Additional comments

I congratulate authors on the valuable addition to the research field. However there is a room from improvement. I suggest to work on more thoroughly in the future directions

Reviewer 2 ·

Basic reporting

The authors have incorporated all the changes listed in first round.

Experimental design

No further changes are required

Validity of the findings

The results are valid and novel.